

# Currents Generated by the Sea Breeze in the Southern Caspian Sea

Mina Masoud[1] and Rich Pawlowicz[2]

[1,2]Dept. of Earth, Ocean and Atmospheric Sciences, University of British Columbia, 2207 Main Mall, Vancouver, B.C.
Canada V6T 1Z4

**Correspondence:** Mina Masoud (mmasoud@eoas.ubc.ca)

**Abstract.** The sea breeze system is the dominant atmospheric forcing at high frequency in the southern Caspian Sea. Here, we describe and interpret current meter observations on the continental margins of the southern Caspian from 2013 to 2014 to identify and characterize the water column's response to the sea breeze system. Time series analysis provides evidence for diurnal baroclinic current signals of $O(0.02 \text{ m s}^{-1})$ and surface height changes of $O(0.03 \text{ m})$. A two-layer model, including

interfacial and bottom friction is developed to further investigate the sea breeze response. This model is able to reproduce the structure, amplitudes, and phases of observed diurnal current fluctuations, explaining half of the variance in observational current response at frequencies at 1 cpd and higher. The sea breeze response thus results in a "tide-like" daily cycle which is actually linked to the local forcing all along the southern Caspian coast.

## 1   Introduction

Diurnal-period onshore/offshore wind variability is a persistent feature of many coastal areas, especially in tropical and sub-tropical areas, but also in temperate zones (Sonu et al., 1973; Simpson, 1994; Steyn, 1998). Due to the smaller thermal heat capacity of land, it heats more rapidly in the day, and cools more rapidly at night, relative to the sea, resulting in land–sea thermal gradients with a daily cycle. This leads to cross-shore pressure gradients which generate onshore/offshore wind flows,

called sea breeze systems, with the same daily periodicity. The diurnal sea breeze system can have a significant impact on the incident wave climate, nearshore processes and morphology in tropical and subtropical regions (Sonu et al., 1973; Masselink and Pattiaratchi, 2001). Coastal currents can also be generated (Hyder et al., 2002; Zhang et al., 2009; Sobarzo et al., 2010; Gallop et al., 2012). These coastal currents can dominate the high-frequency variability over continental shelves (DiMarco et al., 2000; Rippeth et al., 2002; Hyder et al., 2002; Simpson et al., 2002). However, it is often difficult to separate tidal,

inertial, and sea-breeze effects in the coastal ocean response, since the time scales are very similar.

Recently, it was found that the variability of shelf currents in the southern Caspian Sea (Fig. 1a) is mostly dominated by coastally-trapped waves with time scales of several days and longer (Masoud et al., 2019), but that a significant daily signal is also present. The Caspian Sea, about 1030 km long and 310 km wide, is the largest enclosed basin in the world, and the southern coast, which is characterized by a shallow shelf of width 10-30 km with a deeper basin offshore, has a very persistent

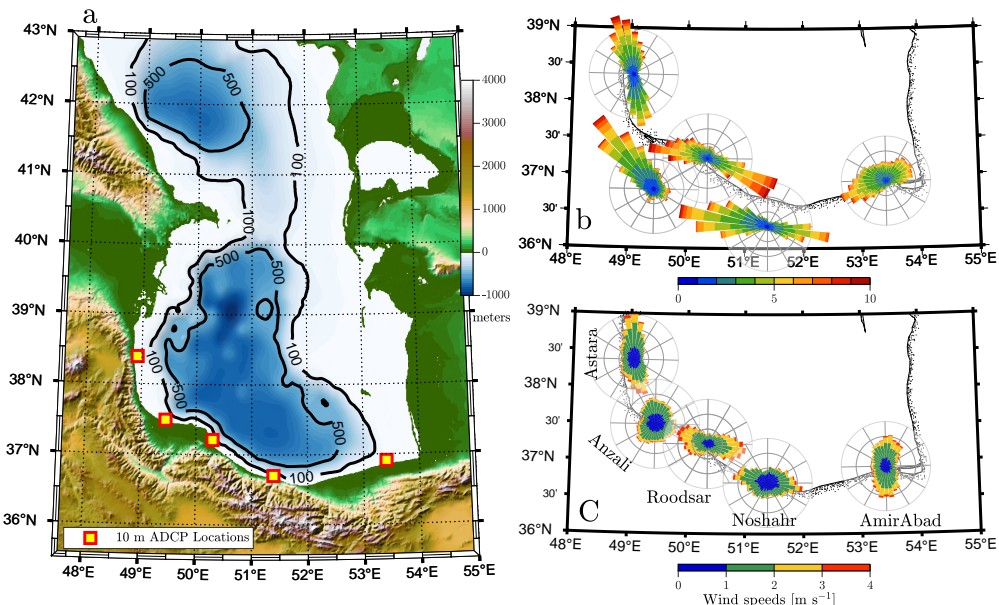

**Figure 1.** a) Southern Caspian Sea with location of current meter moorings. Topography above/below the levels of the Caspian is derived from the ETOPO2 dataset (National Geophysical Data Center, 2006); the water level in the Caspian Sea is 28 meters below mean sea level. b) Wind roses for total winds at mooring locations (Anzali and Noshahr roses are shifted southwards for clarity). c) Wind roses for diurnal band-passed winds. Rings appear at 2, 4, and 6%.

sea-breeze pattern, present through most of the year. Meteorological aspects of this pattern have been well-studied (Khalili, 1971; Khoshhal, 1997; Azizi et al., 2010; Karimi et al., 2016), and analysis of a short current record at one station suggested that this sea breeze was linked to high-frequency variations in water column currents (Ghaffari and Chegini, 2010). Further, since tides in the CS are very weak (Medvedev et al., 2016, 2017, 2020), it is possible that most of the higher frequency variations in coastal currents may be a response to the sea-breeze system.

Here, we take advantage of the strong and persistent sea-breeze forcing in the southern Caspian Sea, and the lack of confounding tidal effects, as well as the availability of measured current records at 5 well-separated locations along the southern Caspian shelf obtained between late 2012 and late 2013, to investigate the nature of a geophysical water column response on a shelf to periodic sea breezes. Analytical solutions to a coupled two-layer rotating wind-driven shallow-water model are compared with observations and show good agreement.

## 1.1 The study area

The Caspian Sea (Fig. 1a) is a terminal basin into which rivers flow but water is only lost by evaporation; its surface is about 28 m below sea level. Although the northern Caspian Sea is very shallow, with depths of less than 50 m, the southern part is characterized by a central region with depths of more than 800 m, bordered on the south and west by a narrow shelf area





(Fig. 1), extending 10-30 km offshore to the 100 m isobath. Inshore of this is a coastal plain of varying width, backed up by
the Alborz mountains with heights of up to 5610 m. On the south-eastern coast, the shelf extends offshore more than 100 km;
the coastal plain there is similarly flat and extends well inland.

Most of the freshwater enters from the Volga River in Russia to the north. There are many small rivers on the southern
(Iranian) coast but together they supply only 5% of the freshwater input (Alizadeh et al., 2008). The large-scale stratification
in the Caspian's water column varies seasonally, with warm salty (20-30°C, 12 PSU) waters in a relatively well-mixed layer
about 40-100 m deep in summer and fresher, less warm (10°C, 11 PSU) surface waters in winter (Zaker et al., 2007), above
more stratified waters at depth. However, even within this mixed layer there is often a weak stratification.

Atmospheric forcing governs the mostly cyclonic mean circulation of the Caspian Sea, but winds are generally weak in the
southern Caspian with mean speeds of only 3-4 m s$^{-1}$; wind speeds are less than 5 m s$^{-1}$ more than 90% of the time (Kosarev,
2005). The occasional strong winds along the southern Caspian coast result in the formation of baroclinic coastally-trapped
waves along the shelf edge (Masoud et al., 2019). These waves propagate from west to east at speeds of 1-3 m s$^{-1}$, and explain
most of the variance in currents at frequencies less than 1 cpd.

The southern Caspian has a humid subtropical climate characterized by warm summers and mild winters, and receives a
significant amount of solar radiation (Kosarev, 2005). At higher frequencies the sea breeze is then an important phenomenon
which exists throughout the year but is most widespread in spring and summer months (Khalili, 1971; Khoshhal, 1997; Azizi
et al., 2010; Ghaffari and Chegini, 2010; Karimi et al., 2016). A typical sea breeze in warm months is generated by solar
radiation. However, in other months when the temperature gradient between the sea and land surfaces is low, strong winds
towards land at sea level can strengthen the sea breeze and generate precipitation. Outflows from the Alborz mountains in
winter, known as Garmesh winds, can also increase temperatures in the coastal plain, generating a sea breeze (Khalili, 1971;
Khoshhal, 1997; Karimi et al., 2016).
A typical sea breeze cycle in the southern Caspian is characterized by onshore winds (the "sea breeze") generally starting
more than two hours after sunrise, around 9 am-noon (see, e.g., Azizi et al., 2010; Ghaffari and Chegini, 2010; Karimi et al.,
2016, all times referred to here are in local summer time which is known as Iran Daylight Time IRDT or UTC+4:30). The wind
direction changes to offshore (the "land breeze") around 4-9 pm. The maximum wind speed of about 4 m s$^{-1}$ occurs during
the sea breeze between noon and 4 pm, after the time of maximum temperature gradient between sea and land. The strongest
and most frequent sea breeze days occur in areas around AmirAbad and Anzali where the coastal plain is widest, and the least
sea breeze days are observed around Noshahr and Astara (Azizi et al., 2010; Karimi et al., 2016).

### 1.1.1 Data and data processing

The wind and current meter datasets used here were fully described in Masoud et al. (2019) and only brief details are given here.
Over a period of about 16 months from late 2012 to early 2014, current velocity measurements using 600 kHz Nortek AWAC
Acoustic Doppler Current Profilers (ADCPs) were collected at five locations a few kilometers offshore in depths of about
10 m over the southern Caspian shelf, in successive monthly deployments (Fig. 1 and Table 1). Measurements for the period
of December 2012 to December 2013, when spatial and temporal coverage was most complete, are used here. More sparse





**Table 1.** Location, water depth, and distance to shore for all mooring locations. Also given are the direction of the principal axis of current variations. The details of ADCP located at deeper water are presented in brackets for Astara and Roodsar.

| Station | Astara | | Anzali | Roodsar | | Noshahr | AmirAbad |
|---|---|---|---|---|---|---|---|
| Longitude (°E) | 48.92 | 49.05 | 49.45 | 50.30 | 50.35 | 51.39 | 53.41 |
| Latitude (°N) | 38.39 | 38.37 | 37.49 | 37.21 | 37.23 | 36.70 | 36.91 |
| Distance from shore (km) | 4.2 | 15.8 | 1.3 | 2.4 | 8.9 | 1.5 | 6.4 |
| Water depth (m) | 10 | 31 | 10.5 | 10 | 32 | 10.5 | 13.7 |
| Direction major axis (degrees) | 167.00 | 175.74 | 93.78 | 153.10 | 158.54 | 85.14 | 75.6 |

**Table 2.** Ratio of diurnal variance to high-frequency variance (frequencies higher than 1cpd) for alongshore and cross-shore wind stress and bottom current.

| Station | Astara | Anzali | Roodsar | Noshahr | AmirAbad |
|---|---|---|---|---|---|
| Alongshore wind stress | 64.48 | 66.01 | 68.66 | 64.58 | 60.35 |
| Cross-shore wind stress | 64.76 | 65.76 | 69.73 | 69.02 | 72.20 |
| Alongshore current | 38.88 | 29.47 | 44.74 | 39.69 | 35.43 |
| Cross-shore current | 34.87 | 26.96 | 40.34 | 32.12 | 39.17 |

information is also available at two locations further offshore near the 30 m isobath at Astara and Roodsar. The instruments used collected data every 10 min with a vertical bin resolution of 0.5 m; the lowest useful bin, which we use to show bottom currents, is 2 m above the bottom.

Winds at 10 m above the water surface are extracted from a Weather Research and Forecasting (WRF) model configured for the Caspian Sea region, interpolated to the location of ADCP measurement stations. The WRF model, described at length in Bohluly et al. (2018), is configured with two nests. The $42 \times 52$ outer domain has a resolution of $0.3°$ and the $94 \times 124$ inner domain grid has a resolution of $0.1°$. The 6-hourly ERA-Interim reanalysis data of the European Centre for Medium-Range Weather Forecasts (ECMWF) are used as initial and boundary conditions. The model is run daily starting 6 pm for 1.25 days with 6 hours of spin-up time that is discarded. Bohluly et al. (2018) finds reasonable agreement between modelled and observed winds at several offshore and nearshore buoys. The wind stress is calculated from the 10 m elevation wind velocity from the WRF model using drag coefficients from Large and Pond (1981). Since the WRF model is run on a daily cycle, the diurnal peaks in the spectra that we describe later could be numerical artifacts. However, similarly strong diurnal peaks are observed in spectra of observed wind from buoy data at Anzali and AmirAbad, and from land stations located near Astara, Anzali, Noshahr and AmirAbad stations (not shown), and so we believe the WRF outputs reflect real conditions. In addition, the daily analysis we perform in this paper starts at midnight, and hence any systematic forecast-to-forecast step would occur at figure boundaries.




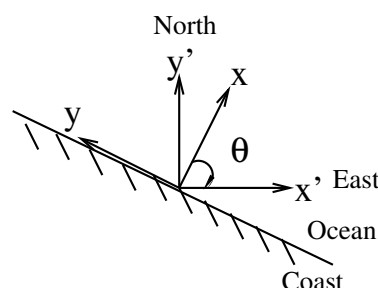

**Figure 2.** Axes definition. Offshore direction is the positive $x$ axis; $\theta$ is the rotation angle between geographic and coastal axes.

Finally, water levels in the southern Caspian are measured by tide gauges at Anzali (37.48°N, 49.46°E) and at AmirAbad
(36.85°N, 53.37°E). For our purposes (to see daily variations) we subtract the daily mean from each day. This removes any
biases resulting from a seasonal cycle with a range of about 0.4 m, as well as long-term trends.

## 2 Results

### 2.1 Total and diurnal-period winds

Wind roses at our 5 study sites (Fig. 1b) show that winds are generally aligned along the coast, with maximum wind speeds of
about 10 m s$^{-1}$. However, if we separate out the diurnal variability using a Butterworth 4th-order band-pass filter to remove
periods less than 6 hours and more than 30 hours, wind roses for this band-limited time series show mostly cross-shelf variation
with speeds of up to 4 m s$^{-1}$ at 3 stations. Diurnal winds at the other two (Astara and Noshahr) still have a significant along-
shore component (Fig. 1c). Astara is at the southern end of a large inland plain from which sea breezes are generated, so that
the sea breeze will align with the coast, and the coastal plain is also very narrow at Noshahr making it difficult to generate a
large cross-shore wind.

Subtracting the mean, the wind stress and current data are then rotated based on principal axes of the currents at 4 m to align
with the local bathymetry so that vectors are decomposed into alongshore and cross-shore components (Table 1, Fig. 2). The
diurnal wind stress represents about 60-72% of the high frequency wind stress variability, depending on location (Table 2) and
the diurnal current variability represents about 27-45% of the high frequency current variability near the bottom.

### 2.1.1 Wind and current spectra

Wind stress spectra have a narrow, statistically significant spectral peak at 1 cpd at all stations (Fig. 3a, c); it is strongest at
Roodsar. Daily energy appears at both positive and negative frequencies, with more energy at negative frequencies everywhere
except at AmirAbad, consistent with the strong directionality of the daily wind rose there (Fig. 1c) and generally clockwise
rotations elsewhere. Smaller spectral peaks also occur at the first harmonic of the diurnal frequency (frequencies of $\pm 2$ cpd) at
many stations, and sometimes (e.g., at AmirAbad) at higher harmonics as well.





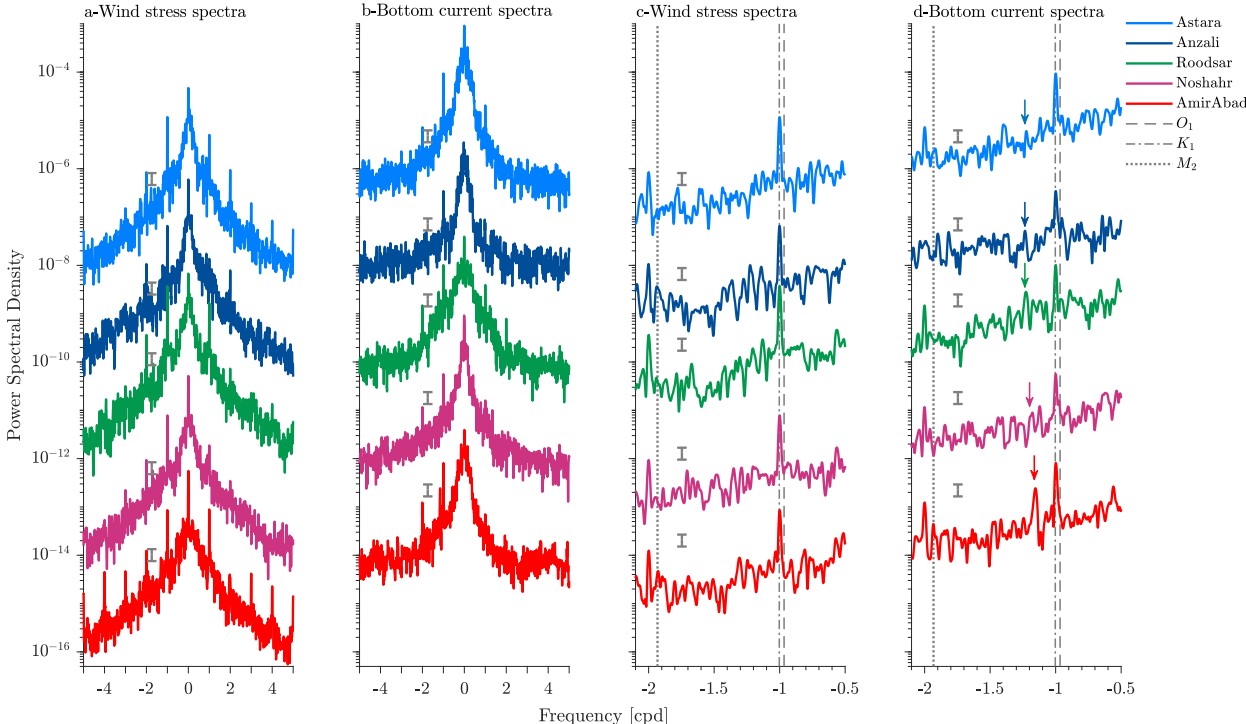

**Figure 3.** Rotary power spectral density estimates of a) wind stress and b) bottom current at Astara, Anzali, Roodsar, Noshahr and AmirAbad stations using the Welch method. Successive spectra are offset downwards by 100 N m$^2$ cpd$^{-1}$ for wind stress and 100 m$^2$ s$^{-2}$ cpd$^{-1}$ for currents. Negative and positive frequencies correspond to clockwise and counterclockwise rotation respectively. The grey error bars indicate 95% confidence intervals. c) A magnification of the wind stress spectra for clockwise diurnal frequencies. d) The same for bottom currents. The arrows indicate the inertial frequency (at approximately 1.2 cpd) at each station, and other vertical grey lines mark the location of the $O_1$ (0.9295 cpd), $K_1$ (1.0027 cpd), and $M_2$ (1.9323 cpd) tidal frequencies.

Rotary spectra for bottom currents also have narrow, statistically significant peaks at 1 cpd, and small peaks at the first harmonic frequencies (Fig. 3b, d). Spectra for wind and currents computed for each season rather than for the whole year (not shown here) also contain the 1 cpd peak. Although these peaks are always present they are largest in the summer and spring. At frequencies higher than about 2 cpd, wind stress spectra continue to slope downwards, whereas bottom current spectra begin

115 to flatten. This suggests that the current time series is mostly dominated by instrument white noise at these high frequencies. We shall then restrict our analysis to frequencies less than 2 cpd.

Inertial frequencies, which at around 1.2 cpd at these latitudes are well-separated from the diurnal frequency, are associated with a very weak peak in bottom currents at most locations (Fig. 3d). Although we cannot separate the diurnal peak from any that might be associated with the dominant diurnal tidal constituent ($K_1$), there is clearly no visible peak at the frequency of

120 the next most important diurnal constituent ($O_1$), or at the frequency of the dominant semi-diurnal constituent ($M_2$), strongly





suggesting that the diurnal and semidiurnal peaks represent a response to wind stress forcing at those frequencies, and not tidal variability.

Coherence analysis (not shown) finds a significant coherence between both cross-shore and alongshore winds with bottom current at 1, 1.2, 2 and 3 cpd. However coherence between cross-shore wind with cross-shore current is stronger compared to alongshore current and alongshore winds at 1 cpd and higher.

### 2.1.2 The sea breeze

In order to concentrate our attention on the sea breeze forcing and response, ignoring the low-frequency variability which was discussed in Masoud et al. (2019), we will analyze only band-passed data from now on. Examining a 4 day period typical of the summer (Fig. 4i), the daily cycle of the sea breeze system is obvious, with onshore wind (the sea breeze) in the late morning-early afternoon and offshore wind (the land breeze) in the the night-early morning. Winds rotate in the clockwise direction. The daily cycle of band-passed wind directions for the whole study period demonstrates the predominance of this daily change from onshore to offshore wind (Fig. 5) over the whole year. At all locations the wind blows onshore in the early afternoon, starting at about 5 hours after sunrise (even as the time of sunrise varies over the year), and the onshore direction changes to offshore around sunset, remaining in that direction until late morning (Figs. 4i and 5).

The diurnal bottom currents are slightly less consistent from day to day (Fig. 4ii), but show an onshore current in the mornings, and offshore currents in the evenings. Although these currents also mostly turn clockwise, they are not in phase with the winds, and their magnitude varies over the whole year (not shown) from less than 0.01 m s$^{-1}$ to as much as 0.2 m s$^{-1}$ on occasion.

More quantitatively, we count the number of sea breeze days at all locations using a standard algorithm, applied to our band-passed datasets. In most selection methods for sea breeze days, the diurnal reversal of wind direction from offshore to onshore is used as an identifier for a sea breeze day (Masselink and Pattiaratchi, 2001; Furberg et al., 2002; Miller and Keim, 2003; Azorin-Molina and Chen, 2009). Additionally, a rapid change in the intensity of wind is considered in some cases. Here, a sea breeze day is counted when 1) the wind direction during the day (from 10 am to 11 pm) is from the sea breeze direction (onshore), but the wind overnight (from 11 pm to 10 am) is not from the same direction (has greater than 60 degree wind direction difference); and 2) the wind direction in the afternoon and morning are both from the sea breeze direction (onshore), but afternoon (noon to 11 pm) wind speed is larger than wind speed in the morning (10 am to noon).

Using this selection criteria, about 220-280 sea breeze days occur in 2013 depending on the location, with a mean wind speed of 1.5 m s$^{-1}$ (Fig. 6). Most are seen at Roodsar (which also has the strongest winds), and least at Astara. However, sea breeze activity is subject to a slight seasonal variability. In spring and summer (April-September), approximately 20-30 sea breeze days are experienced every month. However, closer to 10-25 sea breeze days occur per month in fall and winter seasons (October-March).

Water level measurements (Fig. 4iii) are available at two locations. The daily range is about 0.1 m at both. At Amirabad there is a "low" water level around noon, and a high water level a few hours after midnight. There is in addition a larger twice-a-day signal at Anzali. If we perform a tidal harmonic analysis using T_Tide (Pawlowicz et al., 2002), we find $M_2$ tidal amplitudes

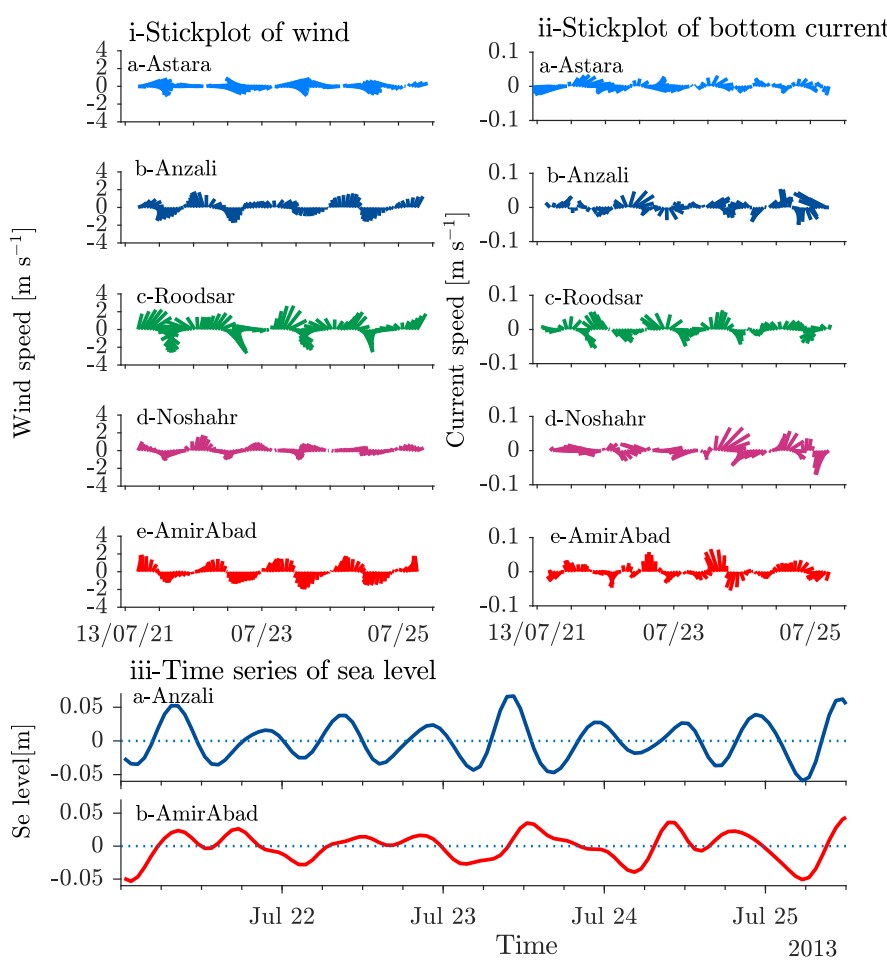

**Figure 4.** An example of summer daily variability from 21-25 July 2013. i) Stick-plot of hourly band-passed alongshore and cross-shore winds at a) Astara, b) Anzali, c) Roodsar, d) Noshahr and e) AmirAbad. In this figure the coastline is horizontal with water above and land below, positive upwards (positive y) winds in the morning are offshore and negative downward (negative y) winds in the afternoon are onshore. ii) Stick-plot of band-passed alongshore and cross-shore current at a) Astara, b) Anzali, c) Roodsar, d) Noshahr and e) AmirAbad. iii) Time series of sea level at a) Anzali and b) AmirAbad.




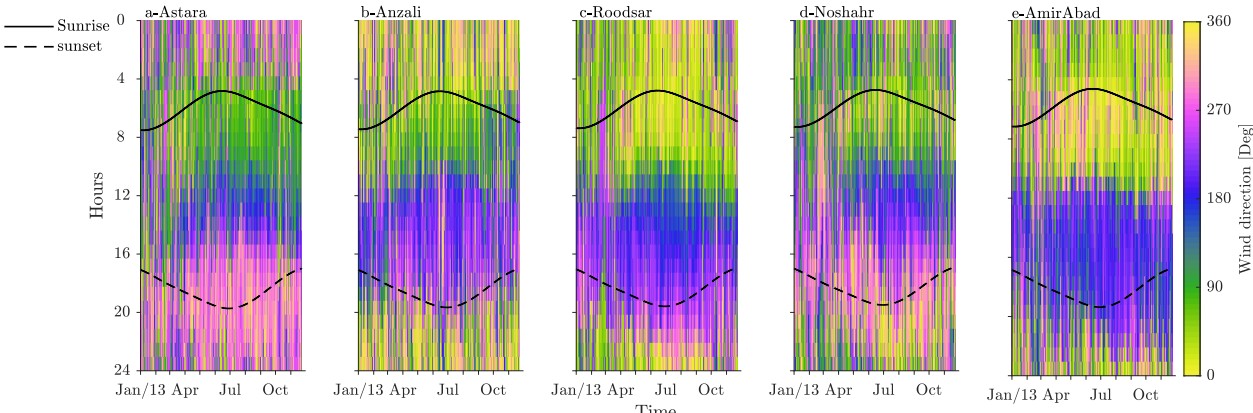

**Figure 5.** Wind direction at a) Astara, b) Anzali, c) Roodsar, d) Noshahr and e) AmirAbad. Angles increase clockwise from the offshore direction, so that a wind direction of $0/360°$ is pure offshore wind and $180°$ is pure onshore wind. The black solid and dashed lines are sunrise time and sunset times respectively (Beauducel, 2001).

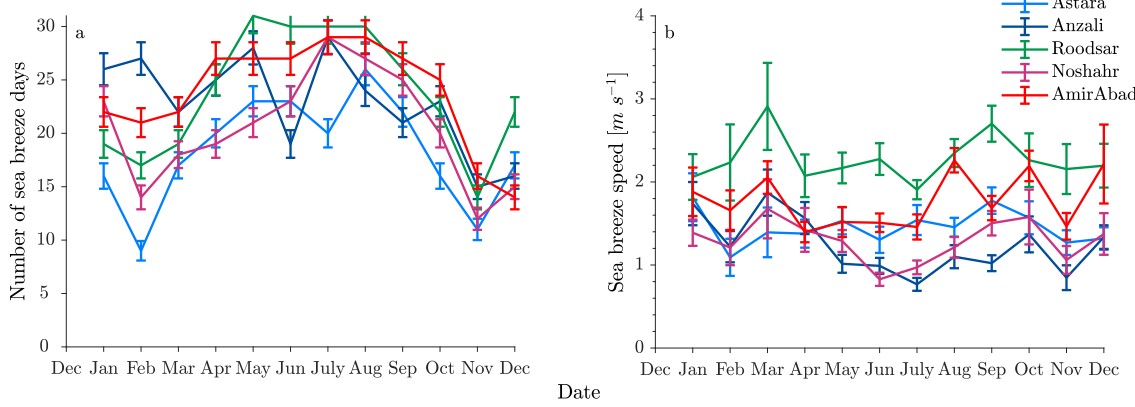

**Figure 6.** a) Number of sea breeze days and b) sea breeze speed at Astara, Anzali, Roodsar, Noshahr and AmirAbad stations for each month from December 2012 to December 2013.

of 0.02 m and 0.007 m for Anzali and AmirAbad stations, respectively. $O_1$ amplitudes are below the noise level, but $S_1$, at 0.01 and 0.03 m for Anzali and AmirAbad respectively, are significantly larger than their close neighbours $P_1$ and $K_1$. $S_2$ is also substantial. However, Medvedev et al. (2016), examining water level records in the central Caspian, suggests that the anomalously large $S_1$ and $S_2$ constituents actually represent "radiational" tides, probably resulting from the sea breeze. The narrowness of these spectral peaks are then a result of the extremely consistent sea breeze pattern over the whole year.



### 2.1.3 The mean diurnal cycle

Now we consider an "average" day. Although the annual changes in sunrise and sunset times result in a slight annual modulation in the timing of the sea breeze (Fig. 5), we ignore this variation and average by hour of the day over the whole year. We also processed the data using only the deduced 'sea breeze' days, but find the smaller number of days in the mean gave more variable results than averaging over the whole year; the sea breeze day selection algorithm appears to be overly conservative.

The resulting time series of daily wind stress (Fig. 7) again shows the same general pattern at all stations, but demonstrates a little more clearly how the magnitude of the signal, and the relative strengths of cross-shore and along-shore wind stresses, varies from place. Along-shore winds are to the right in the morning, and to the left (when facing offshore) in the afternoon and evening. These winds are strongest at Roodsar, and weakest at Anzali. The daily cycle is not a pure sinusoid, but contains distortions associated with higher harmonics. These are greatest at AmirAbad, consistent with appearance of wind spectra

(Fig. 3). In addition, there are more subtle differences in the timing of peaks and transitions. For example, the transition from offshore to onshore flow occurs as early as 9 am at Noshahr, but as late as 11 am at Roodsar. The transition back to offshore flow occurs at 4 pm at Noshahr and Astara, but as late as 11 pm at Roodsar and Amirabad.

In the water column, the average diurnal response is similarly uniform in its patterns at all locations, although the magnitude and exact timing of the response also varies from place to place (Fig. 7). The cross-shore response is almost entirely baroclinic,

with a node at a height above bottom of around 6 to 7.7 m, a short distance above the middle of the water column (Fig. 7iii). Although we do not have measurements close to the bottom or surface due to limitations imposed by the ADCPs design, it seems likely that this pattern consists of the first baroclinic mode, and that surface currents are coherent with, but even larger than, those seen in the topmost bin for which reasonable averages can be obtained. After midnight, there is an offshore flow in the surface layer, apparently matching the offshore wind stress, and onshore flow at the bottom layer. An opposite response

with an onshore flow in the surface layer (and an onshore wind stress) and offshore flow at the bottom layer can be observed during daylight hours. The magnitude of the average response is $O(0.01 \text{ m s}^{-1})$, largest at Roodsar and Astara, and smaller at the other 3 locations. Oscillation peaks, as well as peaks in the offshore wind stress, occur slightly later at Roodsar and AmirAbad relative to the other stations. These delays do not consistently trend eastwards or westwards, and hence do not suggest along-shore propagation of wave-like features.

The alongshore response is also quite similar at all stations (Fig. 7). Here however a noticeable barotropic flow can be seen, in addition to a baroclinic response. Current maximums and minimums of $O(0.01 \text{ m s}^{-1})$ near the bottom lag those at the surface by about 1/4 wave period, so that they reach a maximum while surface values approach zero (and vice versa). In the diurnal alongshore current pattern, there is a negative (rightward) flow in the daytime and positive (leftward) flow in the nighttime.

The stronger winds at Roodsar and Astara are correlated with stronger currents, and weaker winds at Anzali and Noshahr are associated with weaker currents. The timing of changes in the direction of winds and the timing of changes in the direction of currents, which do very slightly from location to location, are also linked; locations with later peaks and zero-crossings in wind





**Figure 7.** The 24 hours-averaged daily cycle of band-passed alongshore and cross-shore wind stress (first panel), alongshore current (second panel) and cross-shore current (third panel) at a) Astara, b) Anzali, c) Roodsar, d) Noshahr and e) Amirabad stations from December 2012 to December 2013. The current data is band-passed and then averaged by hours; we remove values from bins that are too close to the surface and (at Amirabad) bins with unstable averages.





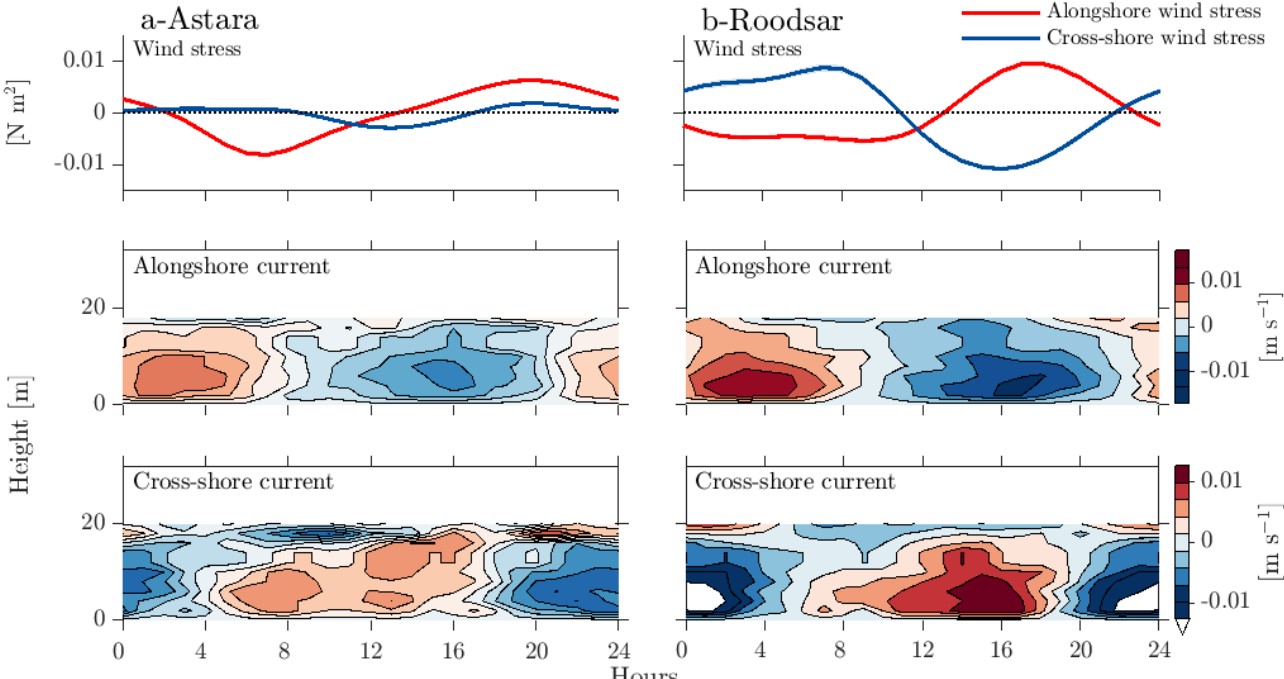

**Figure 8.** The 24 hours-averaged daily cycle of band-passed alongshore wind stress and cross-shore wind stress (first panel), alongshore current (second panel) and cross-shore current (third panel) at a) Astara and b) Roodsar at about 31 m (Tabel 1).

stress time series also have later peaks and zero-crossings in current time series. There is therefore a high (local) correlation between the sea breeze system and the diurnal currents all along the southern Caspian coast.

In addition to the moorings at depths of $\sim 10$ m, two additional current meter moorings at Astara and Roodsar were also located further offshore at the 30 m isobath (Table 1). Although no useful data was returned from the upper half of the water column there, the daily cycle of currents in the lower half (Fig. 8) are similar in direction, magnitude, and timing to that seen at the bottom in the shallower locations, and there also weak indications at the shallowest depth for which reliable measurements can be obtained that an upper layer is present with flows similar to the upper layer flow in shallower waters. Thus daily

oscillations in both surface and bottom waters, with similar pattern and timing, are probably present in the water column over wide areas of the shelf.

## 2.2    Theoretical water column response to the sea breeze

To further understand the linkages between the diurnal surface wind stress and the diurnal currents, we now attempt to model the dynamics. Instead of following the depth-dependent "oscillating Ekman-layer" approach of Craig (1989b) with a vertical

eddy viscosity, coupled to a barotropic mode, which has been used by many authors, we restrict ourselves to a mathematically



simpler coupled two-layer system, as suggested by our observations, for which analytical solutions are more straightforward to obtain.

Thus, consider a linearized two-layer shallow-water model on the semi-infinite plane bounded by a coastline on the $y$-axis, with the positive $x$ axis pointed offshore (Fig. 2) into shelf waters of depth $H_T \approx 10$ m (numerical values for these and

other parameters are presented here without comment to justify the mathematical development; we shall discuss their origin in the next section). Since we are considering a local response over scales of the shelf width, we will filter out long shelf waves (which in any case are not suggested by our observations at daily frequencies) by assuming negligible variation in the alongshore direction, but we retain the possibility of an along-shore wind stress. Also, since the mooring locations are well inshore of the shelf break, and energy that propagates across the shelf break will not return, we can neglect the increase in

depth past the shelfbreak. The upper layer of undisturbed depth $H_1 \approx 3.5 - 6$ m is then governed by

$$u_{1t} - fv_1 = -g\eta_{1x} + \frac{\tau^{(x)}}{\rho H_1} - \frac{r}{H_1}(u_1 - u_2) - Ru_1 \tag{1}$$

$$v_{1t} + fu_1 = \frac{\tau^{(y)}}{\rho H_1} - \frac{r}{H_1}(v_1 - v_2) - Rv_1 \tag{2}$$

$$(\eta_1 - \eta_2)_t + H_1 u_{1x} = 0 \tag{3}$$

and the lower layer of undisturbed depth $H_2 \approx 6.5$ m is governed by:

$$u_{2t} - fv_2 = -g[(1-\varepsilon)\eta_1 + \varepsilon\eta_2]_x - \frac{r}{H_2}(u_2 - u_1) - Ru_2 \tag{4}$$

$$v_{2t} + fu_2 = -\frac{r}{H_2}(v_2 - v_1) - Rv_2 \tag{5}$$

$$\eta_{2t} + H_2 u_{2x} = 0 \tag{6}$$

$$\tag{7}$$

with $g$ the gravitational acceleration, $(u_i, v_i)$ are velocities and $H_i$ are layer depths for the upper ($i = 1$) and lower ($i = 2$)

layers, $\varepsilon = (\rho_2 - \rho_1)/\rho \approx 2 \times 10^{-4}$ with $\rho_i$ layer densities and $\rho$ a reference density, and $\tau^{(x)}$ and $\tau^{(y)}$ applied wind stresses in the offshore and alongshore directions. In this set of equations, $r$ is an interfacial friction, and $R$ represents a bottom friction, both characterized by their time scales $r^{-1}$ and $R^{-1}$ respectively. We (somewhat inconsistently) include $R$ in both the upper and lower layer equations since this allows us to completely separate the baroclinic and barotropic modes next. The equations for each layer are fully coupled by the appearance of the interface height $\eta_2$ in both, as well as by the interfacial friction; wind

stress affects only the surface layer. The coastline boundary condition is that $u_1 = u_2 = 0$ at $x = 0$.

Using procedures described in Section 16 of LeBlond and Mysak (1981), these 6 coupled equations can be approximately separated into two independent sets of 3 equations each when $\varepsilon \ll 1$. A barotropic mode for which

$$u_1 = u_2 \quad \text{and} \quad \eta_1 = \frac{H_1 + H_2}{H_1}\eta_2, \tag{8}$$





implying the two interfaces move together in the same direction with about the same magnitude, and that currents are the same

from top to bottom, is then governed by equations:

$$u_t - fv \;=\; -g\eta_x + \frac{\tau^{(x)}}{\rho H_T} - Ru \tag{9}$$

$$v_t + fu \;=\; \frac{\tau^{(y)}}{\rho H_T} - Rv \tag{10}$$

$$\eta_t + H_T u_x \;=\; 0 \tag{11}$$

where

$$u \;\approx\; \frac{H_1 u_1 + H_2 u_2}{H_T} \tag{12}$$

$$\eta \;\approx\; \eta_1 \tag{13}$$

$$H_T \;\approx\; H_1 + H_2 \tag{14}$$

and thus this mode is mostly linked to sea-surface height variations. The intrinsic speed of high-frequency waves is $\sqrt{gH_T} \approx$ 10 m s$^{-1}$.

In addition, there is also a separate baroclinic mode for which

$$u_2 = -\frac{H_1}{H_2}u_1 \quad \text{and} \quad \eta_1 = -\varepsilon\frac{H_2}{H_1 + H_2}\eta_2, \tag{15}$$

governed by:

$$u_t - fv \;=\; -g\varepsilon\eta_x + \frac{\tau^{(x)}}{\rho H_1} - (\frac{r}{H'} + R)u \tag{16}$$

$$v_t + fu \;=\; \frac{\tau^{(y)}}{\rho H_1} - (\frac{r}{H'} + R)v \tag{17}$$

$$\eta_t + H'u_x \;=\; 0 \tag{18}$$

where

$$u \;\approx\; u_1 - u_2 \tag{19}$$

$$\eta \;\approx\; -\eta_2 \tag{20}$$

$$H' \;\approx\; \frac{H_1 H_2}{H_1 + H_2} \tag{21}$$

which implies that for this mode, velocity shear is linked to mid-water interface depth changes (which are far larger than the

associated surface height changes), and high frequency interface displacements travel with an intrinsic speed of $\sqrt{g\varepsilon H'} \approx$

0.07 m s$^{-1}$, much slower than for the barotropic mode. More importantly, the friction for the baroclinic mode must be greater

than or equal to that affecting the barotropic mode, but the forcing stress is also larger (see changes in denominator of the wind

stress terms).

Now, we wish to find the response of this system to a known diurnally oscillating (and possibly rotating) wind stress.

Fortunately, the equations governing both the barotropic and baroclinic modes are almost identical, albeit with coefficients





whose numerical values are different, and so that the same analytic solution can easily be adapted for either. For simplicity, let us consider a canonical set of equations:

$$
\begin{aligned}
u_t - fv &= -g\eta_x + T^{(x)} - r'u \\
v_t + fu &= T^{(y)} - r'v \\
\eta_t + Hu_x &= 0
\end{aligned}
\tag{22}
$$

and now assume that both alongshore and cross-shore wind stress vectors decay offshore with a length scale $\alpha^{-1}$ ($\approx 100$ km), and are oscillatory with a daily frequency $\omega$ modelled by the real part of:

$$
T^{(x)} = T_0^{(x)} e^{-\alpha x - i\omega t} \quad \text{and} \quad T^{(y)} = T_0^{(y)} e^{-\alpha x - i\omega t}
\tag{23}
$$

for constants $T_0^{(x)}$ and $T_0^{(y)}$.

We look for solutions that vanish as $x \longrightarrow \infty$. The total solution is made of a particular solution to the forced problem and

a homogeneous solution to the unforced equations, which are added together to match the coastal boundary condition. For the particular solution, we guess that $u$, $v$, and $\eta$ will also decay offshore with a scale $\alpha^{-1}$, and oscillate with a frequency $\omega$:

$$
u = U_p e^{-\alpha x - i\omega t}, \quad v = V_p e^{-\alpha x - i\omega t}, \quad \eta = N_p e^{-\alpha x - i\omega t}
\tag{24}
$$

where it is implicit in this approach that we take only the real part of the final (complex) solution. A non-dimensional decay scale, $\sigma = r'/\omega$ (which we will find to be $\approx 0$ for the barotropic mode, but $\sim 1$ for the baroclinic mode), is defined to consider

frictional effects. The particular solution then satisfies:

$$
\begin{aligned}
-i\omega(1+i\sigma)U_p - fV_p &= g\alpha N_p + T_0^{(x)} \tag{25}\\
-i\omega(1+i\sigma)V_p + fU_p &= T_0^{(y)} \tag{26}\\
-i\omega N_p - H\alpha U_p &= 0 \tag{27}
\end{aligned}
$$

whose solution for horizontal velocities in matrix form is

$$
\begin{bmatrix} U_p \\ V_p \end{bmatrix} = \frac{1}{\beta} \cdot \begin{bmatrix} i\omega(1+i\sigma) & -f \\ f & i\omega\left\{(1+i\sigma) + \frac{gH\alpha^2}{\omega^2}\right\} \end{bmatrix} \cdot \begin{bmatrix} T_0^{(x)} \\ T_0^{(y)} \end{bmatrix}
\tag{28}
$$

where

$$
\beta = \omega^2(1+i\sigma)^2 - f^2 + gH\alpha^2(1+i\sigma)
\tag{29}
$$

with the height linked to offshore velocities through

$$
N_p = \frac{iH\alpha}{\omega}U_p
\tag{30}
$$




For the homogeneous problem, a wave-like solution is considered:

$$u = Ue^{ikx-i\omega t}, \quad v = Ve^{ikx-i\omega t}, \quad \eta = Ne^{ikx-i\omega t} \tag{31}$$

leading to a dispersion relation:

$$k^2 = \frac{\omega^2(1+i\sigma) - f^2/(1+i\sigma)}{gH} \tag{32}$$

as well as

$$V = \frac{fU}{i\omega(1+i\sigma)} \tag{33}$$

and

$$N = \frac{kHU}{\omega} \tag{34}$$

Although in the inviscid limit there are free waves propagating offshore (real $k$) for frequencies above the inertial frequency, and evanescent waves (i.e. imaginary $k$ with solutions decaying exponentially in $x$) at lower frequencies, as will be the case in the Caspian, once friction is significant then $k$ will be complex. Here we have decays scales of $Im\{k\}^{-1} \approx 200$ km for the barotropic mode, and around 1 km for the baroclinic mode. We can also (by combining with eq. 29) write

$$\beta = gH(\alpha^2 + k^2)(1 + i\sigma) \tag{35}$$

which shows that $\alpha$ will have little effect on the magnitude of the baroclinic response, although it will be important for the barotropic response, including water level at the coast. Friction will directly affect both, but possibly in a complicated way, since the $1 + i\sigma$ term also appears in the numerator for some terms in eqn. (28).

Adding the particular and homogeneous solutions, and setting $U = -U_p$ to meet the coastal boundary condition $u(x=0) = 0$, the complete response is given by:

$$u = U_p \left\{ e^{-\alpha x} - e^{ikx} \right\} e^{-i\omega t} \tag{36}$$

$$v = V_p \left\{ e^{-\alpha x} - \frac{fU_p}{(i\omega - r')V_p} e^{ikx} \right\} e^{-i\omega t} \tag{37}$$

$$\eta = \frac{iHU_p}{\omega} \left\{ \alpha e^{-\alpha x} + ik e^{ikx} \right\} e^{-i\omega t} \tag{38}$$

with $U_p$ and $V_p$ from eqn. (28).

Note that in the baroclinic case, the expressions within the braces will be dominated by the first term (except very near the coast), so that the baroclinic current magnitude and phase response will be similar everywhere on the shelf, and will be of similar magnitude in both the along and cross-shore directions. However, for the barotropic case, the similarity of $\alpha$ and $Im\{k\}$ mean that the cross-shore barotropic current response will be very small, and may be much smaller than the along-shore barotropic response. These conclusions about relative amplitudes are in general accord with our observations (Fig. 8 ).
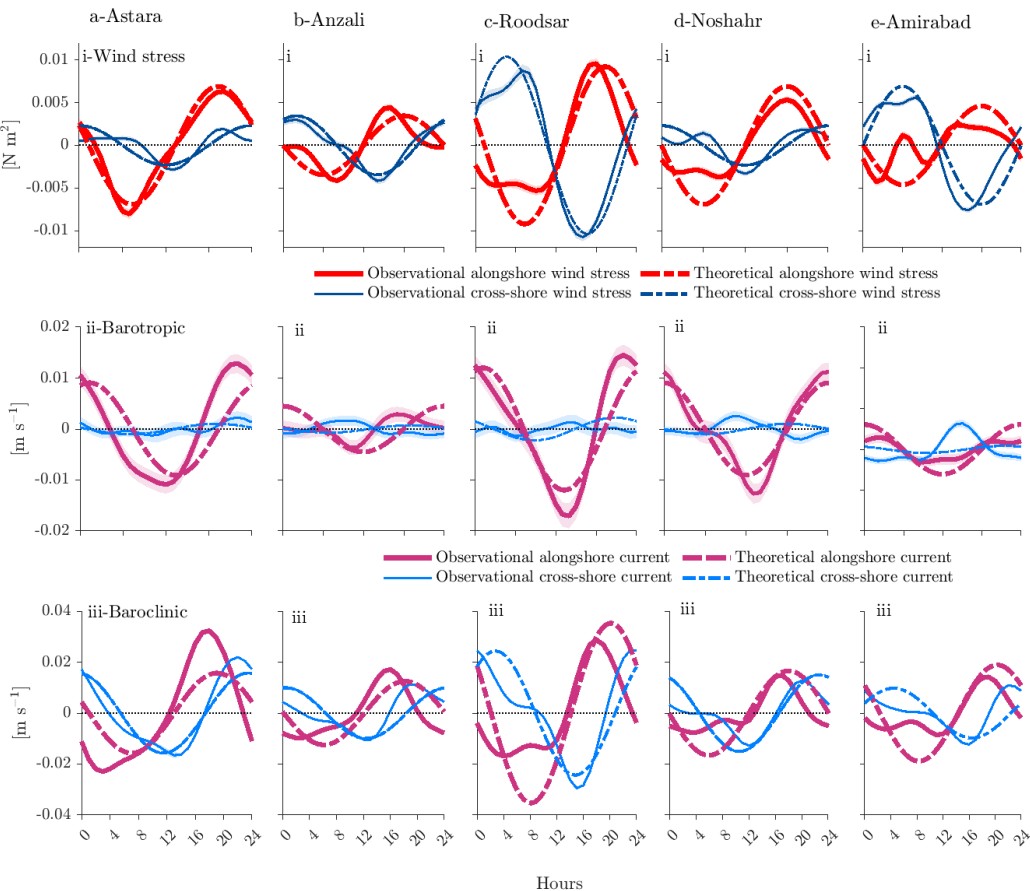

**Figure 9.** Mean daily cycle of i) observed (solid line) and theoretical (dash-dot line) alongshore and cross-shore wind stress, ii) observed (solid line) and theoretical (dash-dot line) barotropic alongshore and cross-shore current response and iii) observed (solid line) and theoretical (dash-dot line) alongshore and cross-shore baroclinic current response at a) Astara, b) Anzali, c) Roodsar, d) Noshahr and e) AmirAbad stations. The shading indicates 95% confidence level for the observed current data.

From our observations, we have both cross-shore and along-shore winds of similar magnitude, and in this case it becomes difficult to generalize further about the relationships between currents and the wind stress. Thus, for further analysis we now try and tune the predicted response to our observations by first matching the measured daily wind cycle (i.e., finding $T_0^{(y)}$ and $T_0^{(y)}$ specifically for each location), and then, taking the offshore distances and layer heights from our observations, adjusting the offshore decay scale $\alpha^{-1}$ and frictions $R$ and $r$ as global parameters. to match the observations.

### 2.2.1 Model/data fitting

Fitting sinusoids with a period of one day to the daily wind stress time series to estimate $T_0^{(y)}$ and $T_0^{(y)}$ for each location is straightforward (Fig. 9i), as these time series are clearly dominated by the daily variations with only a small amount of energy




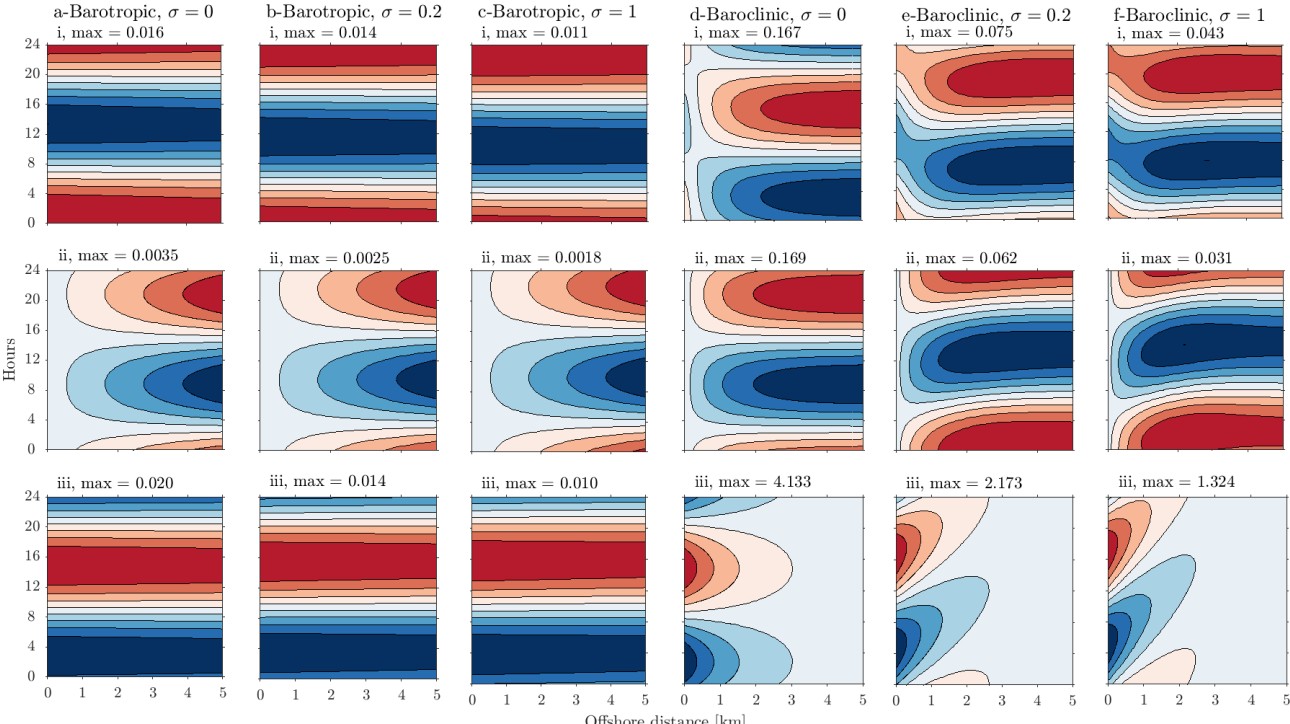

**Figure 10.** Sensitivity of the modeled a) barotropic response with $\sigma = 0$, b) barotropic response with $\sigma = 0.5$, c) barotropic response with $\sigma = 1$, d) baroclinic response with $\sigma = 0$, e) baroclinic response with $\sigma = 0.5$ and f) baroclinic response with $\sigma = 1$ for i) Alongshore current, ii) cross-shore current and iii) sea level with $\alpha = 1/150 \ km^{-1}$ at Roodsar station. The red and blue color represent positive and negative values, respectively.

in the higher harmonics, as we have seen earlier (Fig. 3). However, the lack of ADCP data near the surface results in some difficulty in separating the barotropic and baroclinic modes in the water column observations. The layer interface is evident from the baroclinic response in Fig. 7 at about 6-7.7 m above the bottom at different stations, and this is not centered in the depth range for which observed velocities $\bar{u}_o = (u_o, v_o)$ are available. Using this information as well as the surveyed total water depths (Table 1), we take layer heights in pairs of (4, 6), (3.5, 7), (3.5, 6.5), (4, 6.5) and (6, 7.7) for (surface, bottom) layer

thicknesses at Astara, Anzali, Roodsar, Noshahr and AmirAbad, respectively. Similarly we can take the offshore distances as observed from Table 1.

Next, we estimate the barotropic response by averaging observed current velocities $\bar{u}_o$ at equal distances above and below the apparent layer interface, and the baroclinic response by subtracting current velocity at these depths (Fig. 9ii, iii), For example if the interface was judged to be at 6 m:

$$\bar{u}_{\text{barotropic}} = \frac{\bar{u}_o(8 \text{ m}) + \bar{u}_o(4 \text{ m})}{2}, \qquad \bar{u}_{\text{baroclinic}} = \bar{u}_o(8 \text{ m}) - \bar{u}_o(4 \text{ m}) \qquad (39)$$





The alongshore barotropic response is largest at Astara and Roodsar (Fig. 9ii), where along-shore winds are also largest. The cross-shore barotropic response is small compared to the alongshore barotropic response at all locations. In contrast, the alongshore and cross-shore baroclinic responses are similar in magnitude to each other at all stations (Fig. 9iii), although together they are strongest at Astara and Roodsar; the baroclinic response with peak values of $O(0.02 \ \mathrm{m \ s^{-1}})$ is also about
twice as large as the barotropic response.

Hydrographic profiling did not occur regularly during the current meter program, and although we have found some data its quality is rather low. Nevertheless, it does suggest that there may be a weak stratification over the shelf, and from it we very roughly estimate that the water column is characterized with a nondimensional density difference between layers of $\varepsilon \approx 2 \times 10^{-4}$. Note, however, that the exact value of this parameter is not too important, as its main dynamical effect here
(other than to ensure a baroclinic mode exists) is to set an the offshore decay scale for the effects of the coastal boundary. As long as $\varepsilon$ is small, this response generally occurs only inshore of our mooring locations and hence will not affect the quality of our fits, nor will it have any effect on the response over the rest of the shelf offshore.

The offshore decay scale for the forcing $\alpha^{-1}$ has been estimated to be about 150 km by comparing the energy magnitude in the diurnal peak of wind spectra at different locations offshore perpendicular to Anzali, Noshahr and AmirAbad stations
(1.5 km, 10 km, 40 km, 150 km and 300 km). Changing $\alpha^{-1}$ from 10 km to 300 km has only a small impact on phase shift and magnitude of the modeled baroclinic response, and that mostly near the shelf edge (not shown) as $\alpha x \to 1$. However, the magnitude of the barotropic response (especially for cross-track velocity and the amplitude of surface height changes) does depend directly on $\alpha$ through its importance in the $\beta$ factor (eqn. 29) as was discussed above.

The most sensitive tuning factor is then the friction. However, it too has only a limited ability to modify the solutions. Taking
the phase and magnitude of winds for Roodsar, increasing friction from 0 to $\sigma \approx 1$ decreases the magnitude of the velocities for the barotropic mode (Fig. 10 left side), but causes virtually no difference in the phase of the barotropic cross-shore velocity. Increasing friction does result in a slight advance in phase of the along-shore velocity, and a slight delay in the phase of the surface height cycle. Its largest effect is in greatly decreasing the magnitude of the surface height change, halving it for $\sigma \approx 1$. Note that the along-shore velocity is similar at all locations across the shelf, and the cross-shore velocity is small but increases
linearly with distance from the coast.

The baroclinic mode, on the other hand, has a velocity response which is far more sensitive to friction. Increasing friction to $\sigma = 1$ reduces the velocity magnitudes to about 1/4 of the inviscid values, and significantly delays the phase by almost a quarter cycle. For very weak friction, both phase and amplitude are affected, but with larger $\sigma \sim 1$ the major effect is to reduce the amplitude. Along- and cross-shore velocities have similar magnitudes everywhere offshore. However, there are significant
changes in both the velocity amplitudes, phases, and interface heights very near the coast.

The barotropic response at all locations is then quite adequately matched by an inviscid barotropic mode (Fig. 9ii). Note that the barotropic response actually rotates counter-clockwise, although this is difficult to see since the current ellipse is so narrow. The observed baroclinic response, on the other hand, is somewhat delayed relative to the inviscid solutions, and friction of $\sigma = 1$ must be added to both capture this delay, and match the observed amplitudes. The baroclinic response rotates in a
clockwise direction.





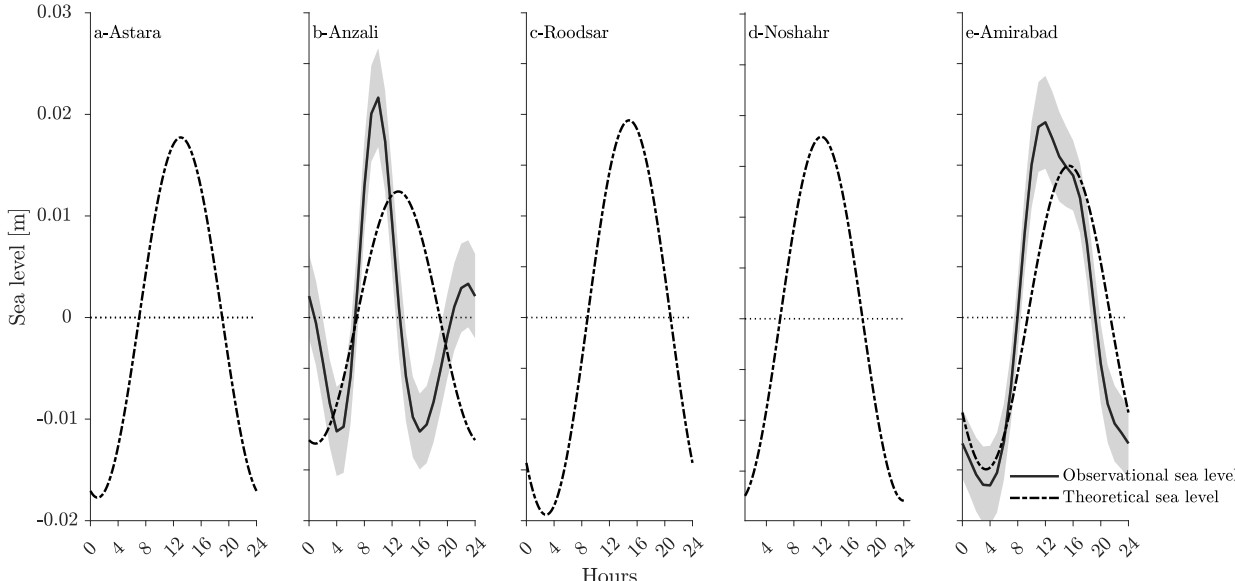

**Figure 11.** Mean daily cycle of observed water level (solid line) only at Anzali and AmirAbad stations and theoretical (dash-dot line) water level at a) Astara, b) Anzali, c) Roodsar, d) Noshahr and e) AmirAbad. The shading indicates 95% confidence level.

Given the limited amount of tuning possible, the predicted responses are, in general, quite close to our observations in both amplitude and phase. In particular, the predicted amplitude and phase of barotropic alongshore current and baroclinic alongshore and cross-shore current are in a reasonable agreement with the observations at all stations. However, the observed responses sometimes contain large departures from a daily sinusoid.

Finally, our model can also provide estimates of layer height changes. Observations of surface height are available at the coast near two of our locations. The observed "mean daily" cycle at these locations shows a range of about 0.02 m (Fig. 11) at all locations. Both the amplitude and phase of the sea-breeze-forced response are in reasonable agreement with these observations. Predicted mid-water interface height changes related to the baroclinic mode at our mooring locations offshore are actually slightly smaller than the surface height changes, making them difficult to discern in our observations since the vertical bin size

in our measurements was 0.5 m.

## 3    Discussion

In the southern Caspian Sea, the sea breeze system, with winds of up to 4 m s$^{-1}$ and wind stresses of up to about 0.02 N m$^{-2}$ (but on average peaking at 2 m s$^{-1}$ and less than 0.01 N m$^{-2}$) is the major diurnal-inertial period process in the atmosphere, explaining about two-thirds of high-frequency variance in winds (Table 2) and hence dominating the forcing of coastal pro-

cesses at high frequency. Ghaffari and Chegini (2010) found that these winds were highly correlated with currents in the high





frequency range at a mooring west of AmirAbad and speculated that there was a link between the sea breeze system and high frequency variance in currents. Anomalously large $S_1$ constituents in tidal analyses for coastal Caspian Sea water levels also suggest a noticeable radiational effect which has in the past been ascribed to sea breezes (Medvedev et al., 2016). We find here that daily-frequency current variations of $\pm 0.02$ m s$^{-1}$ and daily surface height changes of about $0.03$ m are clearly consistent

385 with the water column response to the local sea breeze forcing all along the southern Caspian coast. In both the along-shore and cross-shore directions this daily response is large and baroclinic; there is also an along-shore barotropic component which is about half as large, and an even smaller cross-shelf barotropic component linked to coastal water level changes.

 In comparison, lower-frequency coastally-trapped waves in the same area, generated by lower frequency wind variations, are associated with rather larger barotropic current variations over the shelf (although they can have depth structure further

390 offshore), of $O(0.1$ m s$^{-1})$, mostly in the along-shelf direction, and surface height changes of $O(0.1$ m$)$, which are also larger than those for sea breeze (Masoud et al., 2019). Known processes that affect water level also include an annual cycle of magnitude $O(0.4$ m$)$ due to seasonal imbalances between river inflow and evaporation, and astonomically-forced tidal signals as large as $0.06$ m (Medvedev et al., 2020). Thus, the water column response to the sea breeze does not dominate time series of currents, although it is clearly an important factor in short-term coastal water level changes, resulting in a small "tide-like"

395 daily cycle. The sea breeze current response is, however, very consistent over time, and so is also clearly visible as distinct peaks in spectra of the currents (Fig. 3) and sea level (not shown), as well as in time series in which the lower-frequency motions are filtered away (Fig. 4).

 If we require a band-passed wind speed to be greater than a particular threshold to be classified as a true sea breeze, as is typically done in sea breeze studies (Masselink and Pattiaratchi, 2001; Furberg et al., 2002; Miller and Keim, 2003; Azorin-

400 Molina and Chen, 2009), we count more than 220 sea breeze days in 2013, with an average wind speed of $1.5$ m s$^{-1}$ (Fig. 6). These details change from location to location, however, with average speeds at Roodsar about double the speeds at Noshahr and Anzali. In summer, more than 20 sea breezes occur monthly. Less frequent sea breeze days are observed in winter. On the other hand, time series of band-pass filtered wind angle (Fig. 5) show that this daily reversal is present at nearly all times, although again the details of timing change from location to location.

405 Our findings for 2013 thus agree with earlier work in concluding that the sea breeze is an obvious and featured phenomenon in the southern Caspian Sea area, especially in spring and summer months (Khalili, 1971; Khoshhal, 1997; Azizi et al., 2010; Ghaffari and Chegini, 2010; Karimi et al., 2016). Also in agreement with this earlier work, we find that the sea breeze starts in late morning sometime between 9-11 am (depending on time of year and location), reaches its maximum velocity between about 12-4 pm, and subsides between 4-10 pm, after which it is replaced by the land breeze (Figs. 4, 5 and 7i). This pattern

410 of onshore sea breeze during the day followed by offshore winds at night particularly in spring and summer is a characteristic feature of many other coastal areas (Rosenfeld, 1988; DiMarco et al., 2000; Simpson et al., 2002; Hyder et al., 2002; Zhang et al., 2009; Sobarzo et al., 2010; Gallop et al., 2012), however most of these other investigations were based on observations from only one or two (usually close together) locations. One significant result here is that the sea breeze system in the southern Caspian Sea and its water column response is shown to be coupled in a very similar way over a distance of about 500 km

415 along a coastline, although the exact timing of both the forcing and the response does vary from place to place. The variations





are not consistent with a phase propagation along the coast, as was found at lower frequencies where the eastward delays were associated with the passage of coastally-trapped waves (Masoud et al., 2019). This suggests that the response to the widespread diurnal forcing is mostly local, which would be consistent with the evanescent (i.e., non "wave-like") nature of an oceanic forced response, as we are north of the critical latitude for daily variations.

Not only is a sea breeze seen along the coastline, but the offshore extent of the sea breeze was estimated here to be 150 km. Although the typical scale of sea breeze system is 50 km for subtropical areas described by Sonu et al. (1973); Simpson (1994) and Steyn (1998), other studies (Largier and Boyd, 2001; Simpson et al., 2002) reported the existence of strong diurnal oscillations on the outer Namibian shelf, and at Benguela shelf edge. Significant diurnal winds extending to an offshore site of 125 km from the coast of north Africa at latitude 22°N were reported (Halpern, 1977). Since the shelf area is only 10-30 km
wide, the sea breeze thus affects the entire width of the shelf, and in turn the pattern of currents that arise in response to the sea breeze might also be expected to be similar over the shelf width, except perhaps very close to the coast where the response must adjust to the coastal "wall".

In detail, the local water column response to the sea breeze can be described as follows: in the cross-shore direction, currents are baroclinic with a zero crossing near the middle of the water column (Fig. 7). The sea breeze forces onshore surface flow
and offshore bottom flow during daytime, with the opposite at night. The total excursion for water parcels would be around 600 m. There is little barotropic offshore flow, although it cannot be zero since a small daily variation in surface height is seen at the coast, with lowest waters during daylight hours. Note that the baroclinic response is also weak enough that the mid-water interface also does not vary in height very much over most of the shelf. In contrast, the daily alongshore response is much more strongly barotropic, especially at Roodsar, with flow leftwards at night and rightwards (when facing offshore) in daytime, with
a total excursion of about 300 m.

A diurnal current response to diurnal wind stress with a combined barotropic/baroclinic response in the alongshore direction and two-layer baroclinic structure in the cross-shore direction was also clearly observed on the Chilean shelf at 36-37 °S (Sobarzo et al., 2010). The flow in the surface layer was downwind, and the motion at the bottom layer was in the opposite direction. A baroclinic response has also been seen elsewhere at or poleward of the critical latitude (Hyder et al., 2002; Simpson
et al., 2002; Rippeth et al., 2002; Sobarzo et al., 2010). A unique aspect of our observations is that diurnal bottom and surface currents have almost the same amplitude. In contrast many previous studies have found that the diurnal surface current is stronger than the bottom current, and the amplitude of the oscillations in the bottom layer are weaker than in the surface layer, by a factor which is a function of the depth of the pycnocline (Rosenfeld, 1988; Rippeth et al., 2002; Hyder et al., 2002; Zhang et al., 2009; Sobarzo et al., 2010; Gallop et al., 2012). Here the stratification is weak enough that no distinct pycnocline exists
on the shelf; instead the first baroclinic mode separates the water column column into two almost equal layers.

Theoretical models of the circular motion and vertical structure of current response to diurnal winds in shallow and deep water in the presence of a coast, after the decay of transients, have been investigated by Craig (1989a, b). A one-dimensional, constant-density analytical model was applied with an assumption that the bottom layer flow is forced by the coast-normal pressure gradient due to the periodic wind stress, without considering frictional effects (Craig, 1989b). The mean flow is driven
by a barotropic surface slope resulting from the applied wind stress. This pressure gradient is anti-phase to the surface wind-





stress and so transfers the forcing to the whole water column with a 180° phase shift that leads to anti-phase motions in the lower layers. Simpson et al. (2002) examined numerical solutions including the effects of frictional coupling between layers, extending Craig's approach. Craig's analytical model also has been extended in a numerical solution to a multi-layer structure with frictional coupling between the layers via an eddy viscosity by Rippeth et al. (2002). In this model energy propagates
down through the water column through frictional coupling of adjacent layers as in the classical Ekman problem. However, Rippeth et al. (2002) also used a two-layer analytical model without considering friction effects to demonstrate current response to wind in the diurnal band.

Here, our observations clearly suggest a two-layer response, without along-shore propagation of long waves, which is superimposed on lower-frequency variations arising from coastally-trapped waves, and so we have developed a linear two-layer
model, including interfacial and bottom friction but without along-shore variations, to investigate the sea breeze response and the important factors governing this response in a more general way. Analytical solutions for this forced system greatly simplify the identification of important factors in the response, without making prior judgements about their importance.

For the barotropic mode, bottom friction can reduce the magnitude of the response but it has only minor effects on the phase. The magnitude of the response is also affected by $\alpha$, so to some extent changes in one can compensate for changes in the
other. However, friction must remain weak as increasing friction advances the phase of the (strong) along-shore component and hence would reduce the agreement with the phase of observations. Previous modelling of low-frequency coastally-trapped waves in the southern Caspian (Masoud et al., 2019) required a linear bottom friction of $1.5\times10^{-4}$ m s$^{-1}$ to agree with observations. Spread out over the 10 m water column to agree with our formulation for friction this would be equivalent to $R = 1.5\times10^{-5}$ s$^{-1}$, which, for daily variations, would imply $\sigma \sim 0.2$, larger than any value that we might wish to use here.
It is likely then that this large value represents scattering losses for along-shore wave propagation, rather than truly frictional effects.

For the baroclinic mode, phase effects are larger, and an interfacial friction of $\sigma = 1$ is clearly important for matching the observed baroclinic response. Similarly, Hyder et al. (2002) inferred a relatively large linear friction coefficient ($\sigma \approx 0.6$) to match the phase difference between the alongshore and cross-shore current components. Note that the coastal boundary
condition, whose importance was noted by Chen and Xie (1997) in their numerical model of the sea breeze response on the Texas-Louisiana shelf, is important over the whole shelf for the barotropic model, but is only important very close to the coast for the baroclinic mode.

One obvious problem with our model is that we assume a flat bottom, with a coastal wall, and we change the depth of that flat bottom for our different locations. One result is that the baroclinic response in particular turns out to be quite complicated
in phase and amplitude close to the coast. Instead, the shelf area might be better thought of as a (much more mathematically inconvenient) wedge shape. One could imagine modifying our solutions by approximating this wedge as a series of steps, with two layers in each, along with coupling conditions between "flat-bottom" solutions at each increase in depth. However, since our solutions are evanescent and locally forced, it seems unlikely that such a more complex model will produce qualitatively different solutions. Amplitudes and phases of the response might vary slightly from those derived using the flat-bottom model,
but these could be "tuned" with the friction parameters. A set of moorings extending across the shelf, along with concurrent





density profiling, would be necessary to improve our understanding of the cross-shelf structure; such a program need only extend over a period of about a month in summer since the sea breeze ini the southern Caspian is so consistent.

Although our observations are poleward of the critical latitude where $f = \omega$, the theoretical model we have developed can be equally well applied to locations equatorward of this point. Since an offshore decay scale $\alpha^{-1}$ and friction are included,

the model can also be applied at the critical latitude without difficulty; although in this case the amplitude of the baroclinic mode could be much larger, but also strongly dependent on the magnitude of friction and $\alpha$. Most previous studies have investigated the diurnal current response at the critical latitude ($30°$ N/S). Here the tips of the vector current response trace out almost circular paths, with much larger currents (0.3-0.6 m s$^{-1}$) than we have observed in our study area (DiMarco et al., 2000; Hyder et al., 2002; Simpson et al., 2002; Zhang et al., 2009), the barotropic response is then perhaps not as obvious.

Equatorward of the critical latitude, the offshore response would consist of waves propagating seaward, although we are not aware of any published observations that demonstrate this behavior. Note that the sea breeze response is inherently local, as shown by the evanescent response of the model, and by that strong local correlations between the phase and magnitude of wind forcing with the phase and magnitude of the ocean response. Daily wind variations can affect surprisingly large parts of continental coastlines (e.g., Huang et al., 2010) and so daily ocean responses, perhaps confused with tidal effects, may be

more widespread than currently thought.

The modelled ratio between the maximum cross-shore and alongshore baroclinic currents $v/u$ is $f/\omega$ ($\approx 1.12$ in the southern Caspian) for pure sea breezes in our theory. The current response at the equator would thus tend to be ellipses perpendicular to the coastline, and at the poles, ellipses aligned with the coastline. Our baroclinic velocities do roughly follow this pattern, with $v$ amplitudes slightly larger than $u$ amplitudes. Rosenfeld (1988), Rippeth et al. (2002) and Sobarzo et al. (2010) investigated

a current response to sea breeze poleward of the critical latitude (around $36$-$40°$ N/S, similar to the latitude our case study) and found clockwise and anticlockwise near circular diurnal motions in Northern and southern hemispheres, respectively. They observed the diurnal currents with amplitudes of up to 0.2 m s$^{-1}$ as a response to wind stress in range of 0.1-0.2 N m$^{-2}$ (wind speed of 5 m s$^{-1}$) somewhat larger than our observations but in the same proportion. However, our barotropic velocities, especially at Astara, Roodsar, and Noshahr, have very much larger along-shore components. This is a consequence of the

strong along-shore components of the daily wind variation, likely due to topographic effects inland, which can greatly increase $v$ relative to $u$ near the coast.

One surprising aspect of the analysis is the lack of energy at inertial scales on the shelf. This is particularly true for surface currents (not shown), but spectra for bottom layer currents also show little inertial energy. Consistent with this result, weak near-inertial motions are predicted near coasts of the Caspian basin with depths of less than about 20 m, with maximums in

the energy of near-inertial motion instead occurring in the center of Caspian basin (Farley Nicholls et al., 2012). This suggests that sea breeze forcing is not the source of this near-inertial response. However due to lack of measurement data, further investigation about the source of these strong inertial currents is recommended.

Finally, strongly frictional anti-phase flow in surface and bottom layers could lead to enhanced dissipation and vertical mixing through the water column in shelf areas located close to the critical latitude with significant diurnal winds (Simpson

et al., 2002). Although we are some distance away from this critical latitude, the sea breeze response in the southern Caspian



accounts for nearly half of high-frequency variance and may be the dominant mechanism driving baroclinicity, and thus, in turn, may be the major source energy driving vertical mixing in this area. This is an area which requires further investigation.

*Code and data availability.* Current measurements and sea level data are available upon request from Iranian Ports and Maritime Organization (http://marinedata.pmo.ir). The modeled wind data and the post-processing and analysis scripts are available at https://github.com/mina-masoud.


*Author contributions.* MM performed the analysis and drafted the manuscript. RP developed the mathematical model. Both authors contributed equally to the development of the research concept and the manuscript beyond the initial draft.

*Competing interests.* No competing interests are present

*Acknowledgements.* We thank the Iranian Ports and Maritime Organization for providing the current meter and sea level measurements. We are grateful to Seyed Mostafa Siadatmousavi (IUST) and Masoud Montazeri Namin for giving us his insights about the Caspian Sea. Support provided by The Natural Sciences and Engineering Research Council of Canada under Discovery Grant 2016-03783, Collaborative Research and Development Grant 486139-15, and Metro Vancouver.



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
