# Peer review of "Currents Generated by the Sea Breeze in the Southern Caspian Sea"

_Ocean Science, 2021_

## Referee Comment (RC1)

Review of OS-2021-57 "*Currents Generated by the Sea Breeze in the Southern Caspian Sea*" by Mina Masoud and Rich Pawlowicz

In general, this is an interesting and useful paper, but some of the results can be presented in a better, more "visible" way!

(1) The main comment and suggestion is to calculate and present the  $S_1$  ellipses!

I was very surprized that this was not done, taking into account that the second author is the main author of the world-known program of harmonic analysis of tides! The >1 year-long series of observed currents allow to separate  $S_1$  currents from  $K_1$  and  $P_1$  currents (e.g. Zaytsev et al., 2010). Besides, in fact,  $K_1$ ,  $O_1$  and  $P_1$  tides in the southern Caspian Sea are negligible (Medvedev et al., 2017).

Such constructed S1 ellipses would give lots of important information and enable the authors:

(a) To see the main properties of the  $S_1$  currents, in particular, the amplitude, direction of propagation relative to the coastline and the exact phase;

(b) To see the vertical structure of the  $S_1$  currents and the effects of the baroclinicity;

(c) To effectively compare  $S_1$  currents observed at various stations;

(d) To compare observed and modeled currents in a visible way.

So, this analysis would make the paper much more interesting and understandable for readers! (see Zaytsev et al., 2010 as an example). BTW, it appears that the authors themselves understand the importance of the  $S_1$  ellipses and discuss this question in the end of their Discussion (Lines 501-504).

(2) Section 2.1.1, Figure 3, rotary spectral analysis.

This type of analysis is almost senseless here because stations are located close to the coast and currents are nearly rectilinear. It would be much more useful to show spectra of cross-shore and along-shore components (with the two spectra in one plot for better comparison). Such spectra would give much more helpful information! In fact, several following figures (Figures 7, 8, 9 and 10) are done just for cross-shore and alongshore current components.

Minor comments:

Abstract, Line 2: "from 2013 to 2014" → Throughout the text it is written "December 2012 to December 2013". Be consistent!

Line 37: "...28 m below sea level"  $\rightarrow$  28 m below mean ocean level.

- Line 506: "...found clockwise and anticlockwise **near circular** diurnal motions in Northern and southern hemispheres"  $\rightarrow$  In the open sea, but certainly not near the coast!
- Line 512: "One surprising aspect of the analysis is the lack of energy at inertial scales on the shelf." → There are lots of papers on inertial currents in the Caspian Sea. Actually, they are quite intense in this sea, but... in the open sea, not near the coast!

Line 514: "maximums"  $\rightarrow$  maxima (Latin word!)

**References**

- Pattiaratchi, Ch., Hegge, B., Gould, J., Elliot, I. (1997), Impact of sea-breeze on nearshore and foreshore processes in southwestern Australia. *Continental Shelf Research 17*(13), 1539–1560.
- Zaytsev, O., Rabinovich, A.B., Thomson, R.E. and Silverberg, N. (2010), Intense diurnal surface currents in the Bay of La Paz, Mexico, *Continental Shelf Research 30*, 608-619.

---

## Author Comment (AC1)

**Reply to Reviewers' comments- Currents Generated by the Sea Breeze in the Southern Caspian Sea**
* * *
* * *
**Reply to Reviewers' comments- Currents Generated by the Sea Breeze in the Southern Caspian Sea**

Authors: Masoud, Pawlowicz

**General comments**

We thank the reviewers for their constructive comments. Most of the issues they bring up are relatively minor and were easily addressed, and we go through them all below. However, some larger issues were also raised, which we deal with first.

Reviewer1: The main comment and suggestion is to calculate and present the $S_1$ ellipses. I was very surprised that this was not done, taking into account that the second author is the main author of the world-known program of harmonic analysis of tides! The $> 1$ year-long series of observed currents allow to separate S1 currents from K1 and P1 currents (e.g. Zaytsev et al., 2010). Besides, in fact, K1 , O1 and P1 tides in the southern Caspian Sea are negligible (Medvedev et al., 2017). Such constructed S1 ellipses would give lots of important information and enable the authors: (a) To see the main properties of the S1 currents, in particular, the amplitude, direction of propagation relative to the coastline and the exact phase; (b) To see the vertical structure of the S1 currents and the effects of the baroclinicity; (c) To effectively compare S1 currents observed at various stations; (d) To compare observed and modeled currents in a visible way. So, this analysis would make the paper much more interesting and understandable for readers! (see Zaytsev et al., 2010 as an example). BTW, it appears that the authors themselves understand the importance of the S1 ellipses and discuss this question in the end of their Discussion (Lines 501-504).

We did not perform a harmonic analysis mainly because in the initial stages of the analysis we concentrated on identifying sea breeze days, and analyzing those days alone, and only later found that averaging over the whole year was worthwhile and simpler due the large number of sea breeze days (harmonic analysis for the $S_1$ constituent and its multiples at $S_2$, $S_3$, etc. is roughly equivalent to averaging the daily cycle over the whole year). Of course, just because the initial analysis was done in one way does not mean that another analysis might be better. However, unlike the reviewer, we find the $S_1$ ellipses, which we plot below, somewhat opaque to understanding as they contain both barotropic and first baroclinic mode information, not easily separated. Further, presenting the information at single moorings in $S_1$ ellipse format obscures the relationship between the phase of along and across-shore currents, and time of day (for us at least). Time of day, i.e., morning or afternoon, is generally not important in tidal analysis, but it is clearly important to sea breeze mechanisms.

The only immediately obvious feature of the ellipse figure is that the Greenwich phase does not progress along the coast, but we already discuss this in Lines 180-185 (in contrast, phase does progress along the coast in our earlier coastally-trapped wave analysis for low-frequency variations, published in Masoud et al. (2019). Note that we are poleward of the critical latitude, whereas the area discussed in Zaytsev et al. (2010) is equatorward of this, so we do not expect propagation. Further analysis of vertical phase variations can be used to separate barotropic and baroclinic modes (as was done in Pawlowicz (2002)) but we think the procedure we used here is more straightforward and hence more informative overall, especially since Figures 7 and 9 are more similar to figures that have been used by others reporting on sea breeze and coastal wave problems. An obvious exception to this statement is the analysis in Zaytsev et al. (2010), and if we had known about this paper earlier it might have changed our analysis (we have added this reference at line 499 in revised version). However, we still have the problem that our measurements are missing the upper part of the water column, and so the issues of separating out the baroclinic and barotropic responses, and developing a model to explain them, (which seemed a reachable goal because we are beside "straight coastlines"), occupied more of our time instead of pursuing more complex correlation-type analyses.

Reviewer1: Section 2.1.1, Figure 3, rotary spectral analysis. This type of analysis is almost senseless here because stations are located close to the coast and currents are nearly rectilinear. It would be much more useful to show spectra of cross-shore and along-shore components (with the two spectra in one plot for better comparison). Such spectra would give much more helpful information! In fact, several following figures (Figures 7, 8, 9 and 10) are done just for cross-shore and alongshore current components.

This is a puzzling comment, since the 1 cpd currents and winds are not only NOT rectilinear, but in a number of cases (shown by the assymmetry between positive and negative amplitudes at the 1 cpd peaks in Fig. 3 - the point of using rotary spectrum is specifically to show this assymmetry) are clearly rotational, and this information would be lost in plotting along- and across-coast spectra. Indeed, overplotting the two spectra in a single plot, or alternatively plotting the positive and negative frequencies on the same axes, which is a useful method for comparing spectral levels of the broad-band "background" of red spectral noise, is not actually that informative when trying to see the relative amplitudes of the very narrow $S_1$ peaks (we concede that the figures in Zaytsev et al. (2010), which incidentally also present rotary spectra, do a good job with this).

Moving on to Reviewer 2:

Reviewer2: Overall an interesting set of observations worthy of publication in Ocean Sciences. My only real hesitance is I think the presentation of the work needs to be much more focused on the key results, as at present there is too much unnecessary detail obscuring the key results. In particular, it is not clear to me how the new theory presented differs from previous work on inertial oscillations in coastal settings. What new (generic) incites does this work give to the understanding of inertial oscillations in shallow seas? At present the paper gives the impression that the results are only really relevant to the Caspian Sea which is not the case. As such a recommend the paper to be more clearly focused on the key results (and new incites).

We assume that the reviewers use of the term "inertial oscillation" was a simple mistake, as the daily oscillations we discussed here are clearly NOT at the inertial frequency (see Fig. 3), and in fact we find the level of inertial energy

[Figure]

Figure 1: S1 tidal ellipses at the 5 stations, arranged from westernmost to easternmost. Ellipses are shown by their depth above bottom according to numbers on the vertical axis, but are scaled so that along-shore currents are parallel to the horizontal axis and cross-shore currents are parallel to the vertical axis in the correct aspect ratio . The bar radiating away from the center of each ellipse shows Greenwich Phase.

in these observations extremely low.

Having said that, we are somewhat at a loss here as how to respond. Perhaps the reviewer was hoping for a different paper about sea breezes in general? Being a paper entitled "Currents generated by the Sea Breeze in the southern Caspian Sea" we believe it is focused on the key result, which is that the sea breeze, present most of the time, is directly responsible for about the half of the variance of high-frequency currents in the southern Caspian Sea. Surely this result alone is worthy of being published! We additionally show this by developing a new model linking sea breeze to daily currents, so (at least in the results) it is not clear what details are unnecessary. We do consider inertial oscillations in the Discussion (lines 512-517) but mostly to contrast our weak levels near the coast with the strong inertial oscillations seen offshore.

The reviewer then wonders how the model is new. Lines 202-205 attempted to explain this: "Instead of following the depth-dependent oscillating Ekman-layer approach of Craig (1989b) with a vertical eddy viscosity, coupled to a barotropic mode [...] we restrict ourselves to a mathematically simpler coupled two-layer system [...] for which analytical solutions are easier to obtain." The model has analytical solutions, rather than requiring numerical solution (which makes untangling the relative effects of different forces easier to carry out); and the Ekman layer does not appear in this formulation. Further, adding friction and (possible) wave behaviors means that the model remains valid at all latitudes. Possibly a key difference is that the Craig approach is inherently based on unstratified conditions, although currents are horizontal, whereas our approach is inherently based on the existence of a stratification of some kind to restrict vertical motions. However, it turns out that the magnitude of the stratification, which would be important for (e.g.) determining propagation speeds, is here mostly irrelevant because the response does not propagate as we are poleward of the critical latitude.

The reviewer then asks what generic insights arise from this. We concede that this is a valid question. We have added the following sentences at the end of the paper (lines 526-530 in revised version) to address this comment:

"The observed strong and obvious sea breeze response in the South Caspian Sea which is well-modelled by our two-layer analytical model is likely present in other locations around the world where sea breeze systems exist. However, in those locations the diurnal water response to the sea breeze might easily be overlooked due to strength of tidal fluctuations. Therefore, we suggest more careful examination of motions at the $S_1$ frequency in other study areas. For example a diurnal sea breeze occurs all along the Chinese shelf (Huang et al., 2010) and so this might be a possible location of a widespread oceanic sea-breeze response."

**Detailed replies**

Continuing on to more detailed replies, we have included the original comments from the reviewers and then highlighted our replies by underlining.

*Reviewer 1*

- Page 2..., line ...:Abstract, Line 2: "from 2013 to 2014" Throughout the text it is written "December 2012 to December 2013". Be consistent! Right. It is now changed to late 2012 to late 2013.

- Line 37: "...28 m below sea level" 28 m below mean ocean level. Changed to mean ocean level

- Line 506: "...found clockwise and anticlockwise near circular diurnal motions in Northern and southern hemi-spheres" In the open sea, but certainly not near the coast! Their measurements are on the shelf area, close to the coast. Measurements in Rosenfeld (1988) study are at 30, 60, 90,150 and 400 meter from 5-20 km from the coast in California. Measurements in Rippeth et al. (2002) located at 10-70 km from the coast at 50 to 80 meter depth. Measurement data in Sobarzo et al. (2010) are within 5 km from the coast at around 30 meter depth.

- Line 512: "One surprising aspect of the analysis is the lack of energy at inertial scales on the shelf." There are lots of papers on inertial currents in the Caspian Sea. Actually, they are quite intense in this sea, but... in the open sea, not near the coast!

  Not sure how to respond - not only we are in an agreement with the reviewer's comment, but also this is just what we tried to say. Perhaps this rewording of Lines 516-518 (in revised version) helps?: "Consistent with this result, weak near-inertial motions are predicted near coasts of the Caspian basin with depths of less than about 20 m, but with quite large maximums in the energy of near-inertial motion occurring in the center of Caspian basin (Farley Nicholls et al., 2012)."

- Line 514: "maximums" → maxima (Latin word!) Done!

*0.1. Reviewer 2*

- Line 19: "However, it is often difficult to separate tidal, inertial, and sea-breeze effects in the coastal ocean response, since the time scales are very similar." Is this statement correct? I you need to be more precice?

  We are a little puzzled that this statement is apparently controversial. The inertial frequency is 1.2 cpd, the Sea Breeze frequency is 1.0 cpd and close tidal frequencies are $O_1$ (0.9295 cpd), $K_1$ (1.0027 cpd). These frequencies are so close to each other that they would be difficult to separate in a short time series (say, over a month), especially if the inertial peak is somewhat broad and the sea breeze is intermittent. Of course, opinions about difficulty can vary.

- 43-46: "The large-scale stratification in the Caspian's water column varies seasonally, with warm salty (20-30C, 12 PSU) waters in a relatively well-mixed layer about 40-100 m deep in summer and fresher, less warm (10C, 11 PSU) surface waters in winter (Zaker et al., 2007), above more stratified waters at depth." In terms of the modal structure of the inertial oscillations, the stratification, and it's evolution over the seasonal cycle is key. As such a more accurate description needs to be provided.

  Since we are concerned with current variations in depths of less than 30 m, the details of deeper stratification are not really relevant here (although they were important in our earlier paper on coastally-trapped waves (Masoud et al., 2019). The key sentence is the last one: "However, even within this mixed layer there is often a weak stratification", which again based on data shown in Zaker et al. (2007).

This is because, in terms of the modal vertical structure in these shallow depths, the details of the stratification are actually not that important - the first baroclinic mode will have a zero-crossing near the mid-point of the water column whatever the stratification, as long as there are no sharp steps (which appears to be the case). It is true that the propagation speed of internal modes will be strongly dependent on the magnitude of the stratification, but since we do not have propagating modes for the sea breeze response in this geographic region this effect is unimportant for our paper. The main effect of stratification, other than allowing for a two-layer response at all, is to set the offshore decay scale for the baroclinic adjustment to the coastal boundary condition - but as long as stratification is weak this decay scale is inshore of our mooring locations and so does not require more information (see explanation Lines 336-342). This is very fortunate for us, as there simply is no more information that we know of about seasonal changes in stratification.

Note, however, that if future observational programs include arrays of moorings across the shelf, possibly closer to the shore than this decay scale, better information about the stratification will be required.

- 56: "However, in other months when the temperature gradient between the sea and land surfaces is low, strong winds towards land at sea level can strengthen the sea breeze and generate precipitation." I found this section a little confusing. You need to be clear as to how the sea breeze evolves through the seasonal cycle.

We are simply pointing out that the daily wind cycle is not always forced by the land/sea temperature differences right at the coast. Other processes can also occur. However, although these other processes are important in understanding the mechanisms underlying the daily wind cycle, they are not really important for understanding the coupling to ocean currents.

The mechanisms for generating sea breeze system in the Caspian Sea were investigated by previous studies, and it is perhaps simpler not to try and explain these (sometimes complex) mechanisms here, instead we direct the reader to those references. We therefore reword line 56 as follows:

"However, in other months when the temperature gradient between the sea and land surfaces is low, other mechanisms, for example outflows from the Alborz mountains in winter known as Garmesh winds, can also increase temperatures in the coastal plain, generating a sea breeze (Khalili, 1971; Khoshhal, 1997; Karimi et al., 2016).

- 76: "Weather Research and Forecasting (WRF) model". I key wind characteristic in terms of generation and damping of inertial oscillations is the wind direction. You need to discuss the accuracy of the model wind predictions in these type of coastal situations at some point.

We have added more information about model accuracy, from an additional reference (Ghader, 2014). They carried out a statistical analysis, calculating correlation coefficients, the root mean square error (RMSE), bias and a skill core to evaluate the performance of the WRF model over the Caspian Sea. Based on their results, the estimated RMSE between WRF winds and offshore wind buoy, nearshore wind buoy along the South Caspian Sea (Anzali, Noshahr and AmirAbad) and QuickSCAT satellite data are less than 3.7 for six time periods of

simulation (May 2001, Nov 2006, Aug. 2008, Dec. 2008, Aug-Oct 2011 and Apr-July2013). Over all simulated periods, the cross-correlation and skill core of simulated winds with observed wind data and Quicksat satellite data are mostly higher than 0.6 and 0.7 respectively. We have now added the following information to the text (lines 82-88):

"The accuracy of modelled winds has been evaluated by Bohluly et al. (2018) and Ghader et al. (2014) . The latter compared model winds with a variety of observed wind products over the Caspian Sea including one offshore buoy, three nearshore buoys and also data from the QuikSCAT satellite product. Qualitative and quantitative assessment of these comparisons showed that the simulated surface wind fields are in good agreement with the observational data and QuickSCAT satellite data. We also evaluated the accuracy of the WRF wind data ourselves, comparing with available wind buoy data at 3 locations during 2013 (the wind data is available mostly between May and Sep). The RMSE between WRF wind and observed wind is less than 0.1, 0.11 and 0.2 m s$^{-1}$ at Anzali, Noshahr and AmirAbad respectively. "

We also add the following sentence in line 75:

"Some local observations of surface winds are available at three stations (Anzali, Noshahr and AmirAbad) out of our five stations during 2013, but even this data contains gaps, so for consistency we use winds at 10 m ..."

- 451: "This pressure gradient is anti-phase to the surface wind stress and so transfers the forcing to the whole water column with a 180 phase shift that leads to anti-phase motions in the lower layers." This is not stricktly correct - see Criag (1989). The phase shift is essentially a response to the presence of the coastline and the stratification. This paragraph will be removed.

**References**

Bohluly, A., Esfahani, F.S., Namin, M.M., Chegini, F., 2018. Evaluation of wind induced currents modeling along the Southern Caspian Sea. Continental Shelf Research 153, 50–63.

Ghader, S., Montazeri-Namin, M., Chegini, F., Bohluly, A., 2014. Hindcast of surface wind field over the caspian sea using wrf model, in: Proceedings of the 11th International Conference on Coasts, Ports and Marine Structures ICOPMAS, pp. 24–26.

Huang, W.R., Chan, J.C., Wang, S.Y., 2010. A planetary-scale land–sea breeze circulation in east asia and the western north pacific. Quarterly Journal of the Royal Meteorological Society 136, 1543–1553.

Masoud, M., Pawlowicz, R., Namin, M.M., 2019. Low frequency variations in currents on the southern continental shelf of the Caspian Sea. Dynamics of Atmospheres and Oceans .

Pawlowicz, R., 2002. Observations and linear analysis of sill-generated internal tides and estuarine flow in haro strait. Journal of Geophysical Research: Oceans 107, 9–1.

Rippeth, T.P., Simpson, J.H., Player, R.J., Garcia, M., 2002. Current oscillations in the diurnal-inertial band on the Catalonian shelf in spring. Continental Shelf Research 22, 247–265.

Rosenfeld, L.K., 1988. Diurnal period wind stress and current fluctuations over the continental shelf off northern California. Journal of Geophysical Research: Oceans 93, 2257–2276.

Sobarzo, M., Bravo, L., Moffat, C., 2010. Diurnal-period, wind-forced ocean variability on the inner shelf off Concepción, Chile. Continental Shelf Research 30, 2043–2056.

Zaker, N.H., Ghaffari, P., Jamshidi, S., 2007. Physical study of the southern coastal waters of the Caspian Sea, off Babolsar, Mazandaran in Iran. Journal of Coastal Research 50, 564–569.

Zaytsev, O., Rabinovich, A.B., Thomson, R.E., Silverberg, N., 2010. Intense diurnal surface currents in the bay of la paz, mexico. Continental Shelf Research 30, 608–619.

---

## Referee Report (RR1)

Review of OS-2021-57 "Currents Generated by the Sea Breeze in the Southern Caspian Sea" by Mina Masoud and Rich Pawlowicz

In my previous review I made two important suggestions to the authors:

(1) To provide the harmonic analysis of the S1 currents and show the corresponding ellipses.

(2) To construct the U- and V-spectra (cross-shore and along-shore) instead of rotary spectra, which are not effective close to the shore.

Unfortunately, the authors rejected both of them. Being for many years the Editor-in-Chief (PAGEOPH) myself, I aware that authors have the right to have their own opinion, which can be different from the reviewer's opinion. At the same time, the main job of a reviewer (as I understand it) is to help to the authors, to give them a good piece of advice, to assist the authors in presenting their results in the most spectacular way. As I know quite well, authors frequently cannot look at their own results and presentations of these results from aside ("their eyes are blurred") and from this point of view, an independent advice from the reviewer can be very useful (as an author, many times I had such experience myself).

My suggestions were based on thousands of current velocity series that I processed and presented myself, in particular, on about two hundred for Caspian Sea (although, mostly for the northern and central parts of the sea).

Thus my comments on the authors' replies:

(1) Tidal ellipses.

In fact, the figure constructed by the authors in their reply to the reviewer is quite useful in gives much scientific information! By my opinion, it should be included into the paper. My only suggestion is to show the rotation direction, either by colour or by arrows.

(2) Rotary current spectra.

Figure 3, as it is now, is totally senseless! It is almost impossible (or, at least, extremely difficult) to compare either CW and CCW components for the same depth or spectral changes with depth. I am sorry to say this, but the figure is more appropriate for the Christmas tree than for the authors' scientific paper! The authors refer to the figures by Zaytsev et al. (2010) with shown rotary spectra of breeze-generated currents (Figs. 3 and 5 of that paper),

but in that paper CW and CCW spectra are combined together (instead of a mirror image) and the differences are clearly seen. Moreover, to make the corresponding tidal peaks more evident and to demonstrate their evolution with depth, those authors (including the reviewer) in Fig.5 use the linear X scale.

I would like to emphasize, I do not insist on my comments; I just advise the authors how to show their results in a more spectacular way.

Alexander Rabinovich

---

## Author Response (AR2)

**Reply to Reviewers' comments- Currents Generated by the Sea Breeze in the Southern Caspian Sea**

**Reply to Reviewers' comments- Currents Generated by the Sea Breeze in the Southern Caspian Sea**

Authors: Masoud, Pawlowicz

We thank the reviewers for their constructive comments. Most of the issues they bring up are relatively minor and were easily addressed, and we go through them all below.

**1. Main comments**

This is a comment on the revised manuscript and how the authors responded to the reviewer comments on the original OSD version. The manuscript analyses in detail the sea breeze over the southern Caspian Sea, especially as represented in the WRF model, and the response in currents and sea level especially a few kilometres from the coast, as measured and as represented in an idealised model (two layers, uniform in depth and alongshore). Overall I think there should be minor modifications in respect of the following: Apart from the specific context, publication in Ocean Science should be on account of some new understanding. The text should make this clearer; at present there is much citation of literature with similar findings and any novelty in the present work is somewhat "buried". Perhaps a brief "conclusions" section would help (it could include the new paragraph at the end of the Discussion).

[This and the following comment reinforce the first comment of Reviewer 2. Along with most potential readers, I am not an expert on sea breezes to discern the novelty without help.]

There are sections in the "Discussion" especially which might have been better merged in the Introduction. The Discussion is for discussing the present findings; their relation to the existing literature is certainly in place here but literature cited should be quickly related to the present work. I agree with the authors regarding analysis for S1. As well as the reasons given in their response, "tidal" analysis would impose a sinusoid and so not fully represent a typical daily cycle of the sea breeze. A mean over all days as in the manuscript is preferable. However, the manuscript is vague about how much of the variance is in the daily cycle attributable to sea breeze; see the detailed comment below in respect of Table 2 and associated text. I have no strong view regarding component rotary versus along- and across-shore spectra. I would not have termed the reviewer comment "puzzling" but the authors give reasonable justification for retaining rotary components.

In the main comments, the reviewer suggests:

1- Modifications on the citation: In lines 25, 54, 58 and 413, we cite references related to sea breeze studies in the Caspian Sea including Khalili, 1971,khoshhal, 1997, Azizi, 2010 and Karimi,2016. We removed Khalili,1971 reference, since its results are the same as Khoshhal, 1997. We have added "atmospheric" when we talked about these references

in the discussion to clarify that the findings of these references are about weather system over the Caspian Sea not about the ocean response to the sea breeze system. However, there are other places including lines 17, 19 and 417, that we cite all of studies about ocean response to the sea breeze system in other case studies. Since all of the references in these line are based on observational studies in different case studies and we used all of these references to show their unique finding related to our results throughout the paper, we still think we need to keep them.

2- Emphasising on novelty of our work: We deliberately kept the introduction short as a stylistic preference. However we added a new paragraph in the introduction as follows: We find that the water column response to the sea breeze is measurable and widespread, occurring over the entire southern Caspian shelf year-round. However, the coupling between atmosphere and the ocean is a strongly local phenomenon, with changes in the timing of the daily cycle of currents responding to changes in the timing of the cycle of winds directly overhead, with no sign of propagation effects along-shore (unlike the case for lower frequency current variations). Analytical solutions to a new coupled two-layer rotating wind-driven shallow-water model are compared with observations and show good agreement. Model dynamics also explain the nature of the local response. We also added some words throughout the paper to emphasis on the novelty of our work.

3- Emphasise on the variance of daily cycle: See our explanations in the detailed comments.

**2. Detailed comments**

- Table 1. The caption refers to ". . brackets for Astara and Roodsar" but in fact the details of a deeper ADCP are simply in a second column for these locations. Corrected.

- Table 2. It should be stated in the caption that these "ratios" are percentages. Is this strictly "diurnal variance" or variance in a (narrow?) band around diurnal frequency? The text (lines 101-102) suggests that it may be quite a wide band; the caption should be clear about what is shown. The same question arises again in lines 383-385; exactly what is "two-thirds of high-frequency variance in winds".

  Percentage added in the caption. We modified the caption of table 2 to: Ratio of diurnal variance (band-passed filter with removing periods less than 6 hours and more than 30 hours) to high-frequency variance (frequencies higher than 1cpd) for alongshore and cross-shore wind stress and bottom current. line 384: Since we added an explanation about daily cycle and high frequency in the caption of table 2, we think there is no need to change this sentence in line 384 and it is clear now as we refer to table 2 in this sentence.

- Line 113. You mean "clockwise motion" when you say "negative frequencies" which is best avoided.

  Modified to: There is both clockwise and anti-clockwise motions in diurnal frequencies, although the clockwise motions is stronger everywhere except at AmirAbad ...

- Lines 129-131. This is a little confusing; there are four cases (cross-shore, alongshore wind) x (cross-shore, alongshore current) but they are not discussed systematically.

We removed this part since we did not talk about it later and we did not show any plots or numbers related to this.

- Lines 134, 137. Is "band-passed" as described in lines 101-102? Yes, and we add the following sentence to clarify it more: we will analyze only band passed data (removing data with periods less than 6 hours and more than 30 hours) from now on.

- Lines 136-137. "Winds rotate in the clockwise direction" except at Amir Abad according to figure 3a. In this part, we are talking about 4-day typical data of the summer shown in figure 4, which demonstrates that winds rotate clockwise in AmirAbad in this time period. When we talked about the spectra of whole data shown in figure 3, we mention that AmirAbad is an exception with both clockwise and anti-clockwise rotation.

- Lines 139-140 (for observations) and lines 149-152 (defining sea-breeze days) differ regarding the timings between onshore and offshore; this may decrease the number of sea-breeze days identified. Is this related to the comment on line 170: " the sea breeze day selection algorithm appears to be overly conservative". We define sea breeze days according to the observations in line 140 as follows: At all locations the wind blows onshore in the early afternoon, starting at about 5 hours after sunrise, and the onshore direction changes to offshore around sunset, remaining in that direction until late morning. The sunrise is around 5 am in summer (most of the sea breeze days occur in summer) so 5 after sunrise will be around 10am. So we picked this time as sea breeze direction (onshore direction) in our algorithm. And then in line 150: We pick sea breeze days for the sea breeze selection algorithm as follows: Here, a sea breeze day is counted when 1) the wind direction during the day (from 10 am to 11 pm) is from the sea breeze direction (onshore), but the wind overnight (from 11 pm to 10 am) is not from the same direction and 2) the wind direction in the afternoon and morning are both from the sea breeze direction (onshore), but afternoon (noon to 11 pm) wind speed is larger than wind speed in the morning (10 am to noon). The time of onshore and offshore direction is picked approximately based on figure 5 and it is true that the algorithm is conservative.

- Figure 7 caption, line 4. Do you mean Amir Abad or Noshahr? AmirAbad

- Line 198. "do very" → "vary" Corrected.

- Figure 8 caption, Table 1 and line 202 should agree about the water depth. In caption of figure 8, we added 31 m for Astara and 32 m for Roodsar. same for line 202

- Figure 9. What is the value of sigma used for these plots, c.f. discussion in lines 367-371. We add "($\sigma = 0$)" after "inviscid".

- Figure 10 needs a colour scale. What are the units? Individual panels show time-and-spatial-dependence; "sensitivity" (to sigma) applies overall and should be better explained in the caption. 10b, 10e headings and caption disagree on value of sigma. The color scale is added to figure 10. The units added too. The label for

figure 10b and 10e is corrected now. Also one sentence added in the caption to clarify that this is a sensitivity analysis against $\sigma$.

- Lines 377-378. The "agreement" is not so good at Anzali where the observations have a significant (tidal?) semi-diurnal component. The reviewer is right that the modeled water level is not quite in agreement with the observations in Anzali. So we will mention this in line 380 as follows: Although the amplitude of the modeled water level is in a good agreement with the observed one in Anzali, the phase of the modelled response does not catch the semi-diurnal tidal component in this station.

- Lines 468-469. It is not immediately clear what "one" and "the other" refer to. Please specify. Modified to: The magnitude of the response is also affected by $\alpha$, so to some extent changes in either bottom friction or $\alpha$ can compensate for changes in magnitude of the response.

- Lines 500-501. ". . , and . . response." needs a verb to have meaning. Modified to: Note that the sea breeze response is inherently local, as shown by the evanescent response of the model, and by the existence of strong local correlations between the phase and magnitude of wind forcing with the phase and magnitude of the ocean response.